# Divergent effects of climate change on future groundwater availability in key mid-latitude aquifers

Wen-Ying Wu [1,2], Min-Hui Lo [1✉], Yoshihide Wada [3], James S. Famiglietti[4], John T. Reager [5],
Pat J.-F. Yeh[6], Agnès Ducharne [7] & Zong-Liang Yang [2]

Groundwater provides critical freshwater supply, particularly in dry regions where surface water availability is limited. Climate change impacts on GWS (groundwater storage) could affect the sustainability of freshwater resources. Here, we used a fully-coupled climate model to investigate GWS changes over seven critical aquifers identified as significantly distressed by satellite observations. We assessed the potential climate-driven impacts on GWS changes throughout the 21st century under the business-as-usual scenario (RCP8.5). Results show that the climate-driven impacts on GWS changes do not necessarily reflect the long-term trend in precipitation; instead, the trend may result from enhancement of evapotranspiration, and reduction in snowmelt, which collectively lead to divergent responses of GWS changes across different aquifers. Finally, we compare the climate-driven and anthropogenic pumping impacts. The reduction in GWS is mainly due to the combined impacts of over-pumping and climate effects; however, the contribution of pumping could easily far exceed the natural replenishment.

[1] Department of Atmospheric Sciences, National Taiwan University, Taipei 10617, Taiwan. [2] Department of Geological Sciences, The University of Texas at Austin, Austin, TX 78712, USA. [3] International Institute of Applied Systems Analysis, Laxenburg, Austria. [4] School of Environment and Sustainability and Global Institute for Water Security, University of Saskatchewan, Saskatoon, Canada. [5] NASA Jet Propulsion Laboratory, California Institute of Technology, Pasadena, CA 91109, USA. [6] School of Engineering, Monash University Malaysia, Subang Jaya, Selangor, Malaysia. [7] Sorbonne Université, CNRS, EPHE, UMR 7619 METIS, 4 place Jussieu, 75005 Paris, France. ✉email: minhuilo@ntu.edu.tw

Groundwater, the vast water reserve beneath Earth's surface[1], is an essential resource for humans and ecosystems. Globally, more than one-third of the water used originates from underground[2]. In the mid-latitude arid and semiarid regions lacking sufficient surface water supply from rivers and reservoirs, groundwater is critical for sustaining global ecology and meeting societal needs of drinking water and food production. The demand for groundwater is rapidly increasing with population growth, while climate change is imposing additional stress on water resources[3] and raising the probability of severe drought occurrence[4–6]. Therefore, examining how groundwater storage (GWS) may change in response to both climate-driven and anthropogenic effects is crucial.

Climate change influences groundwater systems in several ways[1,7]. In terms of the hydrological cycle, climate change can affect the amounts of soil infiltration, deeper percolation, and hence groundwater recharge. Also, rising temperature increases evaporative demand over land[8], which limits the amount of water to replenish groundwater. By contrast, the anthropogenic effects on groundwater resources are mainly due to groundwater pumping and the indirect effects of irrigation and land use changes[9].

Most estimates of large-scale GWS changes rely on numerical modeling[10,11]. Previous studies[12–21] utilized the General Circulation Models (GCMs) output data to drive the offline simulations of hydrological models and estimate the changes in global and regional groundwater availability. The uncertainties of climate projections from GCMs can be attributed to internal climate variability, inter-model differences, and scenario uncertainties[22]. The offline simulations driven by GCM data involve uncertainties not only in climate projections but also in the downscaling methods. The identification and quantification of these sources of uncertainties are difficult[15]. The offline simulations use a one-way approach that cannot capture the essential land-atmosphere feedbacks when expanding from the pure hydrology to a more holistic Earth system perspective. Moreover, previous studies[23,24] have shown the feedback mechanisms of groundwater to the atmosphere, emphasizing the necessity and advantages of considering groundwater dynamics in coupled climate modeling.

Rather than using an offline model simulation, we use the fully coupled simulations in the Community Earth System Model—Large Ensemble Project (CESM-LE)[25]. The CESM is a fully coupled climate model, including the land, atmosphere, ice, and ocean components, designed to simulate climate changes with internal climate variability[25]. The large ensemble (30 members used in this study) approach accounts for uncertainties from internal variability within the same climate model. Simulations with the length of thousands of years are critical for assessing the decadal-to-centennial trends of the slow-moving, long-memory processes, such as GWS. CESM-LE has been continuously developed[26], evaluated[27], and broadly used for investigating the terrestrial water cycle[28] and its components, such as snowpack[27,29,30], soil moisture[31], snowmelt runoff[32], and water availability[33].

A physically based groundwater parameterization is embedded in version 4.0 of the Community Land Model (CLM4.0)[34–36], which is the land surface model of CESM. CLM4.0 has been developed and evaluated with the Gravity Recovery and Climate Experiment (GRACE) observations[34] (as shown in Supplementary Fig. 1). By simulating the water table depth, groundwater recharge and discharge, and the interactions with the overlying soils, the groundwater parameterization in CLM4.0 can model the physical dynamics of storage changes in the unconfined aquifer[34], which is an essential part of terrestrial water storage[37,38].

It should be noted that the simple groundwater model in CLM4.0 has its limitations at local-scale processes. However, owing to its coupling with the atmospheric and ocean models, CESM is a suitable tool for resolving large-scale multiple interactions and feedbacks of the groundwater system within the hydrological cycle under a fully coupled Earth system framework[23,39,40].

In this paper, to estimate the future climate-driven GWS changes, we used the simulation output data of CESM-LE (2006−2100) under the RCP 8.5 (Representative Concentration Pathway) high-emission scenario. Without considering the anthropogenic effects of water use and management (such as groundwater pumping, dam impoundment, and water transfer), the CESM-LE only accounts for natural physical processes that act on the water budgets, which allows us to investigate the mechanisms and projections of anthropogenic warming-induced GWS changes.

## Results

**Climate-driven GWS changes in key aquifers.** Our results from CESM simulations demonstrate that the changes in GWS do not necessarily reflect only the long-term trends in precipitation change; instead, they are also associated with enhancement of evapotranspiration, and reduction in snowmelt. Under future warming environments, the amount of spring snowmelt as a vital water resource in the seasonal snow-cover regions decreases, whereas ET (evapotranspiration, the sum of evaporation and transpiration) increases due to larger atmospheric water demand. Rainfall, ET, and snowmelt are three factors act as the primary mechanisms for driving GWS changes (Fig. 1).

We assess the future climate-driven evolution for seven world's largest mid-latitude aquifers (Fig. 2a) selected based on their importance to regional water supplies and food production. These regions have been identified as having experienced severe groundwater depletion during the past decade[2,41–46], mostly due to high water demand for irrigation or other water uses[47].

The Central Valley in California accounts for one-sixth of the irrigated land in the U.S.[45]. During drought periods, groundwater withdrawal is increased to compensate for the reduction in the surface water supply. The result from CESM-LE indicates that, if the effect of pumping is not considered, the basin-wide GWS in this region lacks any long-term trend under the future warmer climate (Fig. 2b) as a consequence of several competing effects.

First, less snowfall with much more rainfall (precipitation partitioning) increases GWS in winter (Supplementary Fig. 3); however, the decreased snowmelt reduces groundwater recharge in early spring. Second, slightly larger ET due to higher temperatures leads to enhanced upward capillary fluxes from the water table[48] to the overlying soil moisture for meeting the higher atmospheric moisture demand. These changes in the seasonality of groundwater recharge and the redistribution of water in soil may become challenging for water management in the future despite the fact that the projected future trend of GWS is not significant.

The Southern Plains, located in the central U.S., is another major agricultural region[49,50]. The CESM-LE projects a significant future declining GWS trend of $23.3 \pm 11.4 \, mm \, dec^{-1}$ through the climate-driven effect (Fig. 2b). Studies have shown that the risk of future summer drought is likely to become unprecedented[49] and groundwater recharge decreases[51] over this region, consistent with our result here. Under future warming, the decreases in both infiltration (Table 1, Supplementary Figs. 4, 13) and spring snowmelt can reduce groundwater recharge, leading to the deeper water table and decreased GWS.

The central-north Middle East (Turkey, Syria, Iraq, and Iran), including the Tigris and Euphrates River Basins, falls under both the transboundary and international water management rules[46].

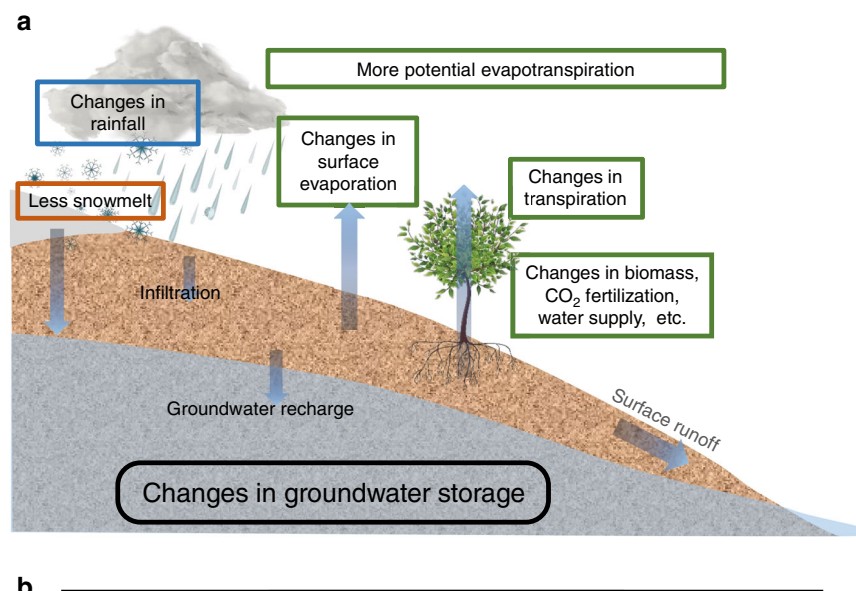

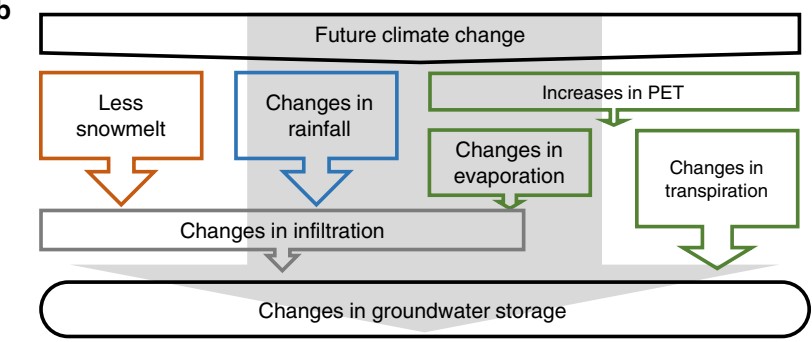

**Fig. 1 Groundwater and climate change. a** Schematic of land hydrological processes. See Supplementary Fig. 20 for each aquifer. **b** Controlling factors affecting divergent groundwater responses. This figure has been designed using resources from www.Freepik.com.

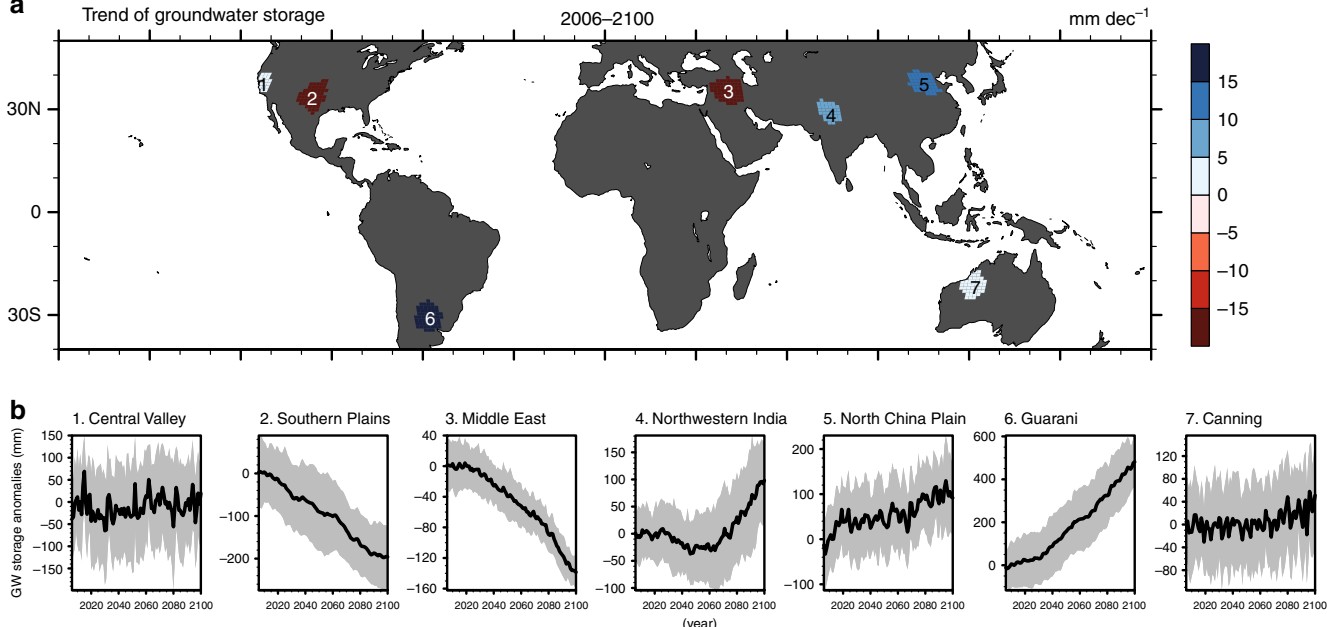

**Fig. 2 Trend of groundwater storage. a** The projected trends (2006−2100) of the climate-driven GWS changes under the RCP8.5 scenario. **b** The corresponding annual time series of GWS anomalies relative to the present (2006−2015) period. The shading denotes the range of one standard deviation among 30 ensemble members.

**Table 1 Trends of future hydrological fluxes and storages.**

| Variable | Unit | 1 Central Valley | 2 Southern Plains | 3 Middle East | 4 Northwestern India | 5 North China Plain | 6 Guarani | 7 Canning Basin |
|---|---|---|---|---|---|---|---|---|
| Total precipitation | mm yr$^{-1}$ dec$^{-1}$ | 10.6 ± 5.9 | 0.7 ± 4.4* | −0.1 ± 2.0* | 6.3 ± 7.7 | 22.8 ± 5.2 | 10.0 ± 4.0 | 9.1 ± 6.7 |
| Rainfall | mm yr$^{-1}$ dec$^{-1}$ | 22.2 ± 5.6 | 5.0 ± 4.2 | 4.3 ± 2.0 | 8.6 ± 7.6 | 26.4 ± 5.2 | 10.0 ± 4.0 | 9.1 ± 6.7 |
| Snowfall | mm yr$^{-1}$ dec$^{-1}$ | −11.6 ± 1.3 | −4.3 ± 0.6 | −4.4 ± 0.4 | −2.3 ± 0.6 | −3.6 ± 0.6 | −0.1 ± 0.0 | 0.0 ± 0.0 |
| Total evapotranspiration | mm yr$^{-1}$ dec$^{-1}$ | 2.9 ± 1.4 | 1.8 ± 3.4 | 1.6 ± 1.7 | 2.5 ± 2.8 | 15.2 ± 1.0 | 6.4 ± 2.6 | 3.9 ± 3.4 |
| Evaporation | mm yr$^{-1}$ dec$^{-1}$ | 4.0 ± 0.9 | 3.9 ± 1.2 | 1.5 ± 1.3 | 4.0 ± 1.0 | 6.4 ± 0.5 | 5.8 ± 0.6 | 6.2 ± 2.5 |
| Transpiration | mm yr$^{-1}$ dec$^{-1}$ | −1.1 ± 0.7 | −2.2 ± 2.5 | 0.1 ± 0.4* | −1.5 ± 2.1 | 8.8 ± 1.0 | 0.6 ± 3.0* | −2.3 ± 1.1 |
| Snowmelt | mm yr$^{-1}$ dec$^{-1}$ | −13.3 ± 1.6 | −4.6 ± 0.6 | −5.6 ± 0.6 | −3.2 ± 0.8 | −3.8 ± 0.7 | −0.1 ± 0.0 | 0.0 ± 0.0 |
| Surface runoff | mm yr$^{-1}$ dec$^{-1}$ | 2.1 ± 1.3 | −1.0 ± 0.6 | −0.5 ± 0.2 | 1.3 ± 1.7 | 3.9 ± 1.4 | 2.8 ± 1.0 | 1.8 ± 1.4 |
| Infiltration | mm yr$^{-1}$ dec$^{-1}$ | 4.5 ± 4.0 | −2.3 ± 3.0 | −0.7 ± 0.7 | 1.0 ± 5.3* | 12.4 ± 3.7 | 1.5 ± 3.5 | 1.2 ± 3.3* |
| Groundwater recharge | mm yr$^{-1}$ dec$^{-1}$ | 5.5 ± 4.1 | −0.2 ± 0.6* | −0.6 ± 0.3 | 2.0 ± 3.5 | 3.7 ± 3.7 | 0.7 ± 0.9 | 3.3 ± 2.9 |
| Groundwater storage | mm dec$^{-1}$ | 1.8 ± 3.3* | −23.3 ± 11.4 | −15.2 ± 3.4 | 7.4 ± 11.1 | 10.0 ± 5.6 | 54.3 ± 20.0 | 3.6 ± 4.3 |

The mean long-term (2006–2100) trends of hydrological fluxes and storages in the seven aquifers estimated from the ensemble mean of the CESM-LE simulations under the RCP8.5 scenarios.
*Asterisk denotes trends that are not significant (p value larger than 0.05).

Groundwater storage in this region is strongly correlated with human exploitation, particularly during drought periods[31]. Our results show that a combination of several interacting factors contributes to the decline in future groundwater recharge (Fig. 2b). At the beginning of this century, groundwater recharge exhibits a strong seasonal cycle peaking in March following snowmelt and infiltration (Supplementary Fig. 5), but this spring groundwater recharge is projected to be markedly reduced at the end of this century (by 77% in MAM). The reason can be attributed to the decrease in snowfall over the Iranian and Anatolia Plateaus, which reduces spring snowmelt, hence infiltration and groundwater recharge. Also, vegetation grows more rapidly owing to $CO_2$ fertilization and higher temperatures, contributing to 13% increased transpiration during the growing season (MAM) and reduced groundwater recharge into the aquifers. Long-term ET increases at a rate of $1.5 \pm 1.3$ mm yr$^{-1}$ dec$^{-1}$, as subsurface water replenishes surface soil via the capillary rise to meet the higher ET demand. Consequently, a continuous decline in GWS ($-15.2 \pm 3.4$ mm dec$^{-1}$) occurs in the Middle East under the warming climate in the twenty-first century (Table 1 and Fig. 2b).

In Northwestern India, the aquifer beneath the upstream regions of Indus River and Ganges River is one of the global aquifers that has experienced the most severe declines in GWS recently[44]. In contrast, our results reveal increasing groundwater resources under twenty-first-century warming when only the climate-driven factors are considered. The projected future long-term GWS trends are associated with rainfall, snowmelt, and ET (60%, 21%, and 19%, respectively, Fig. 3) in Northwestern India. The rainfall increase is the dominant factor, suggesting that without considering pumping effects, the changing climate could provide larger groundwater sustainability in Northwestern India, currently experiencing rapid GWS depletion for supporting irrigated agriculture[44].

Similarly, in the North China Plain[42], a significant increase in precipitation outweighs increasing ET and decreasing snowmelt over the twenty-first century. More precipitation increases infiltration and groundwater recharge (3.7 mm yr$^{-1}$ dec$^{-1}$ in Table 1), which leads to a shallower water table in the North China Plain if pumping is overlooked. However, the seasonal cycle of groundwater recharge shifts (Supplementary Fig. 7), showing changes in groundwater availability across seasons.

Our results also indicate increasing groundwater resources over the Guarani Aquifer in South America and the Canning Basin in Northwestern Australia. Compared to the other five aquifers, these two are characterized by larger annual precipitation (larger than 900 mm yr$^{-1}$), and by a very weak influence of snow. Precipitation is found to be the dominant factor over ET in these two relatively humid regions; thus, increasing P − ET (precipitation minus evapotranspiration) leads to an increase in infiltration and, therefore, in GWS (Fig. 1 and Table 1).

**Mapping climate-driven factors attribution.** The above examples highlight that several physical mechanisms interact to cause different GWS changes globally, either enhancing or competing with each other. To generalize the attribution of future groundwater recharge from the CESM-LE simulations over the twenty-first century, we use the statistical regression with rainfall, ET, and snowmelt changes as predictor variables (see details in "Methods").

Overall, changes in groundwater recharge are dominated by rainfall changes in the monsoon and humid regions (blue colors in Fig. 3). The global regions dominated by snowfall depends on the latitudes and elevation (red colors in Fig. 3). Over dry regions,

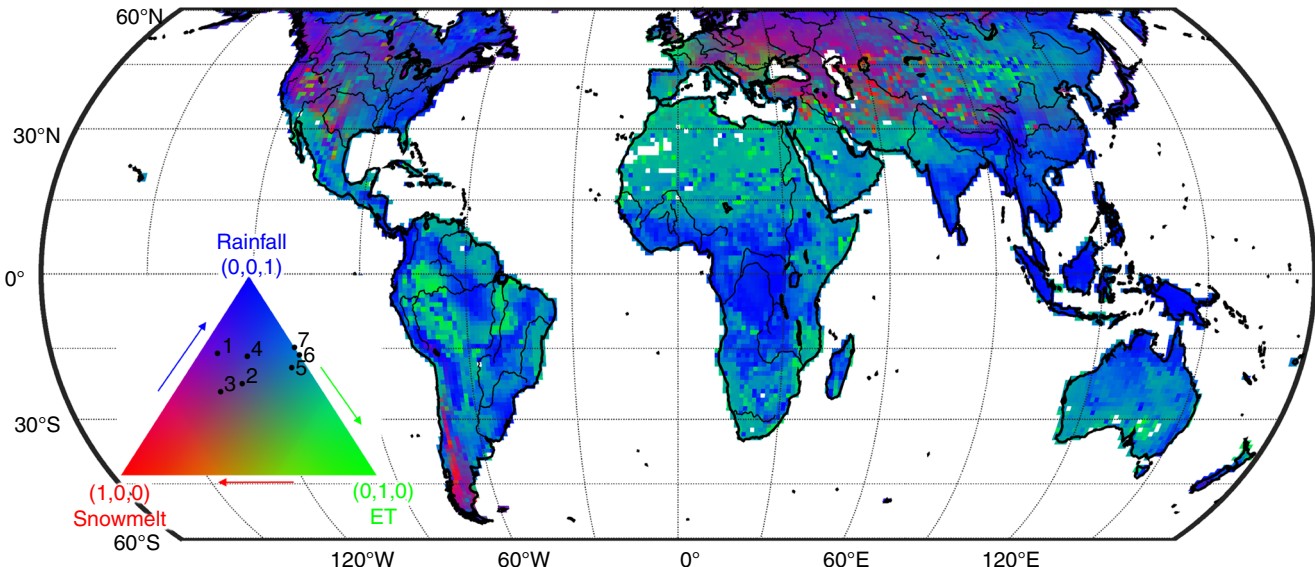

**Fig. 3 Attribution of future groundwater recharge.** Attribution of future groundwater recharge to three climate-driven factors (rainfall, snowmelt, and evapotranspiration (ET)), as derived from the regression based on CESM-LE projections (2006−2010) under the business-as-usual scenario (RCP8.5). The color in the triangle quantifies the relative contribution of each factor based on relative hue in red (snowmelt), green (ET), and blue (rainfall). Aquifer-averaged results are labeled on the triangle using the aquifer index shown in Fig. 2. For example, the results of Guarani (6) is snowmelt: 0.0, ET: 0.4, rainfall:0.6. Three contributions are separated and shown in Supplementary Fig. 11 in a single-color scale.

changes in ET are the dominant factor in groundwater recharge (green colors in Fig. 3).

## Discussion

We explored the potential changes in GWS caused by global warming, using a fully coupled climate model simulation, i.e., land (with an explicit unconfined aquifer representation) + atmosphere + ocean + sea-ice components, which allow physically mechanistic water budget analyses over global domains. The study aquifers are chosen based on the severe decline in GWS during 2003−2014[2,47] (Fig. 4a), which was mainly due to the combined impacts of over-pumping and climate changes[47]. Even though the development of future pumping projections remains difficult and scarce[52], one still can speculate that with the combined negative effects of human-driven depletion and climate-driven replenishment, GWS may continue to diminish globally during the twenty-first century. Here, we provide a comparative assessment of the projected GWS trends of the study aquifers. The satellite-estimated 2003−2014 GWS trends of the climate-driven and the sum of anthropogenic and climate-driven are shown in Fig. 4. A previous study[47] suggests that recent GWS depletion is dominated by the anthropogenic pumping in the Central Valley and Northwestern India. On the other hand, the recent depletion in the Guarani Aquifer and Canning Basin is associated with climate variability, such as a progression from a wetter to a drier or normal period[47]. In Fig. 4b, we compare the results from the "climate-driven" and "climate-driven + anthropogenic pumping" simulations over the twentieth century and the twenty-first-century projection.

The estimated groundwater abstraction was applied to the simulation of the twentieth-century climate with pumping[40] (Supplementary Fig. 2). Generally, significant pumping induces negative GWS trends overtime in the Central Valley, the Southern Plains, Middle East, Northwestern India, and North China Plain (Supplementary Fig. 2). In lightly pumping regions such as the Canning Basin, the GWS trend reflects only natural hydro-climatological drivers. Groundwater depletion over the 2003−2014 period (Fig. 4a) is more significant than the mean rate over the twentieth century (Fig. 4b) due to the accelerating

groundwater withdrawal[40]. In the Southern Plains and the Middle East, the climate-driven groundwater depletion is unprecedented over the twenty-first century ($-18.5$ and $-3.8$ mm dec$^{-1}$, respectively) and far exceeds that over the twentieth century ($-23.3$ and $-15.2$ mm dec$^{-1}$; Figs. 2 and 4b). By comparing the twentieth-century "climate-driven" and "climate-driven + anthropogenic pumping" simulations (in Fig. 4b), we find that the negative effect of pumping to GWS trends can significantly exceed natural replenishment in the heavily pumping regions.

While our explanations of fundamental mechanisms followed a one-way approach, the results shown here considered all known feedbacks in the coupled system. For example, in the Southern Plains, a decrease in soil moisture influences moisture supply for ET, and an increase in rainfall is the result of both regional and atmospheric circulation changes such as the strengthening effect of ENSO teleconnection[53].

A key finding is that future changes in GWS are not only governed by the projected changes in precipitation but also modulated by other hydrological processes (e.g., ET and snowmelt). It is noteworthy that certain mechanisms affecting groundwater recharge in this study are already occurring at present. For instance, changes in timing and the magnitude of spring snowmelt have been reported in observational datasets[54,55] and likely influence the groundwater recharge[56].

Our results also show the large spatial heterogeneity in GWS changes exists in the regions (Supplementary Fig. 10), particularly where the terrain topography is complex, such as Northwestern India and the Central Valley (Supplementary Table 1). Decreasing replenishment from seasonal snowmelt (Table 1) brings negative effects to aquifer storage in snow-dominant regions, a finding consistent with the previous studies addressing the hydrological impacts of climate change in snow-dominant regions[54,57]. Given the relatively coarse resolution of GCMs, we might underestimate the contribution of snowmelt in mountainous regions. It also reveals the necessity to improve the hydrological parameterization and process representation in GCMs for complex terrains such as the downscaling to finer resolutions[58] and the explicit simulation of lateral subsurface flow[59,60].

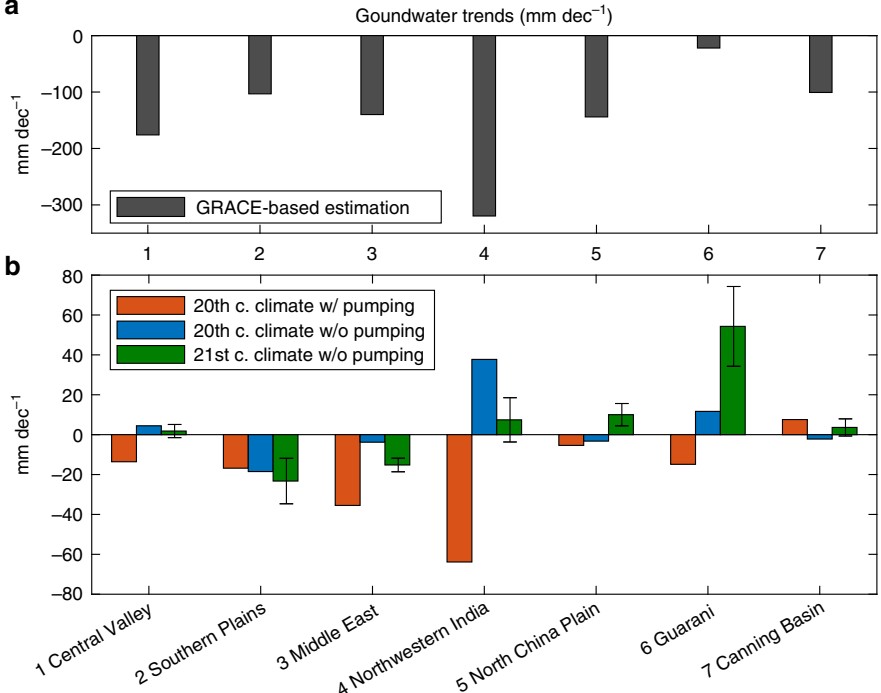

**Fig. 4 Trends of GWS. a** GRACE-based estimates (i.e., GRACE minus GLDAS) of GWS trends (2003−2014). **b** The twentieth-century GWS trends with pumping are estimated based on CESM simulations (1900−1999) in our previous study[40]. The twentieth and twenty-first-century GWS trends without pumping are estimated based on the CESM-LE simulation during 1900−1999 and during 2006−2100, respectively. The error bars represent the spread of one standard deviation among 30 ensemble members.

As a pioneering attempt to address the impacts of climate change on the groundwater budget using a fully coupled Earth System Model, this study is of essential significance toward a better understanding of future changes in the vital but limited groundwater resource. Further investigation and incorporation of future socioeconomic development are necessary to develop the scenarios for future groundwater exploitation rates because the anthropogenic effects are highly likely to exceed the natural climate change effects over global irrigated and urbanized regions[16]. In addition to the three climate-driven factors that we identified in this study, other factors such as changes in soil properties[61], land cover/land use[1], soil freeze-thaw cycles[62], and precipitation intensity[63] may also lead to changes in groundwater recharge and storage in global key aquifers (Supplementary Discussion and Supplementary Figs. 12−18).

Precipitation is known as one of the most dominant factors in all hydrological simulations. However, it is challenging to simulate its spatiotemporal patterns and variability correctly[64,65]. For example, most of the GCMs overestimate the Northern Hemisphere winter meridional winds[66] and, therefore, might overestimate the precipitation in the Central Valley.

To confirm the projected groundwater change is robust in response to anthropogenic climate change, we have analyzed the differences between two 250-year scenarios (Supplementary Fig. 19) and the consensus among the ensemble members (Supplementary Fig. 21). Despite that, our results may remain model-dependent and scenario-dependent. Similar studies but using other fully coupled ESMs and under alternative emission scenarios should be performed to assess better the impacts of global warming on globally important groundwater aquifers.

In closing, although our result highlights the projected increase in future groundwater stress in the Southern Plains and Middle East regions, it also indicates the future possibility of more sustainable groundwater utilization in the Central Valley,

Northwestern India, and the North China Plain where the depletion rates are currently among the highest in the world[2]. In these global groundwater mining hotspots, increasing recharge and changes in groundwater management practices could potentially lead to a substantial slowing of depletion or even aquifer recovery. California's Sustainable Groundwater Management Act is an example of one such ongoing efforts.

## Methods

**Seven key aquifers.** In this study, seven aquifers were characterized by rapid rates of depletion by using observations from the GRACE mission[2]. A previous study showed that water storage declines rapidly during 2002−2013[2]. We use the same boundaries as in a previous study[2] to define the aquifers (Fig. 2a). Information for aquifers is listed in Supplementary Table 1.

**Climate-driven GWS changes.** We used monthly simulations of the Community Earth System Model Large Ensemble (CESM-LE)[25] experiments, performed with a fully coupled land−ocean−atmosphere configuration of CESM, including the Community Land Model (version 4.0; CLM4.0), Community Atmosphere Model version 5, Parallel Ocean Program (version 2), and Los Alamos Sea Ice Model (version 4). CLM4.0 is a global land model established by the Land Model Working Group at the National Center for Atmospheric Research[35,36]. Many land surface processes are well parameterized in CLM4.0, including hydrology, energy, biogeochemistry, and biogeophysics. There is no water exchange between grid cells (columns) in CLM4.0. In this study, we focused on the hydrological processes, including an unconfined aquifer model and a simple TOPMODEL-based runoff parameterization[67]. Changes in GWS result from the changes in groundwater recharge and baseflow. Recharge is represented as the sum of (downward) soil gravity drainage and (upward) capillary rise; a positive value represents water inflows into GWS from the overlying soil layers. Baseflow drains water from the aquifers to rivers. All water fluxes (precipitation, snowmelt, and ET) can potentially affect recharge, baseflow, and GWS through dynamic water exchanges between the land surface, soil moisture, and groundwater. While most CMIP5 models do not include groundwater, the CLM4.0 in the CESM includes an unconfined aquifer to simulate groundwater hydrology. Note that the computations of water states and fluxes in CLM4 are mass conservative, which aids in examining the changes in hydrology from each process.

Thirty different ensemble members, all simulated from the CESM-LE but perturbing in their initial temperature at the level of round-off error in 1920.

CESM-LE includes transient land cover changes (Supplementary Figs. 12–18), but without soil texture changes. All 30 ensemble members are subject to the same model physics, identical external forcing, and the horizontal resolution (0.9° latitude by 1.25° longitude).

The ensemble mean of 30 members was used to diagnose the long-term trend of annual mean GWS, where the shading denotes the one-standard-deviation ranges among 30 members (Fig. 2b and Supplementary Figs. 3–9). The anomalies in Fig. 2b are relative to the mean of 2006–2015. Besides long-term trends, we also analyzed the changes in the seasonality of the water cycles. The changes are calculated based on the mean of 2071–2100 relative to the mean of 2006–2035.

**The regression method for trends**. Before computing the linear trends for a time series (Table 1, Figs. 2, 4), the monthly data are averaged to yearly data. The statistical significance of the trends is from the Student's $t$ test with a null hypothesis that the trend is 0. The null hypothesis is rejected when the $p$ value is larger than 0.05. The uncertainty was quantified by the standard deviation among 30 ensemble members.

**The regression method for contributions**. In Fig. 3, the impacts of three climate-driven factors (rainfall, snowmelt, ET) on natural groundwater recharge is calculated via the following equations[68,69]:

$$y = a_1 x_1 + a_2 x_2 + a_3 x_3 + b, \tag{1}$$

where $y$ is the standardized (subtracting the mean and dividing by its standard deviation) annual groundwater recharge, and $x_i$ is the corresponding standardized annual rainfall, snowmelt, and ET at each global grid (note that $y$ and $x_i$ are not the trends). Only the regions with the statistical significance at a 90% confidence level of multiple linear regression are shown in Fig. 3. The contributions of three climate-driven factors to groundwater recharge is quantified by the following coefficient ranging between 0 and 1:

$$\text{Contribution}_i = \frac{|a_i|}{\sum_{i=1}^{3} |a_i|}. \tag{2}$$

The results are plotted as RGB (Red: snowmelt, Green: ET, Blue: rainfall) for triplet color in Fig. 3. For example, for the Canning Basin (aquifer 7), they are 0, 0.36, and 0.64, and for the Southern Plains (aquifer 2), they are 0.30, 0.24, and 0.46, respectively. $R^2$ for the regression model and the contribution for each factor (the splitting channel) is shown in Supplementary Fig. 11 with a single-color scale.

**Estimating GWS changes from satellite datasets**. For estimating "true" and total (climate-driven and anthropogenic) GWS in the past decade, recent releases of GRACE data[70] from the NASA Jet Propulsion Laboratory with mascon solution and the NASA Global Land Data Assimilation System 1.0 (GLDAS) were used (Fig. 4a). GLDAS forces several offline land surface models with the same atmospheric forcing from observation-based datasets to simulate the water and energy cycle fluxes. To estimate GWS, we used the total water storage from GRACE and removed the storages of canopy water, snow, and soil moisture from GLDAS (multimodel average). This method is commonly used with GRACE-related studies[42–44,46]. The trends are calculated after removing the seasonal.

**Climate-driven and anthropogenic-pumping GWS changes**. To compare the climate-driven and the anthropogenic-pumping effects, we use the twentieth-century CESM simulation from a previous study[40] (orange bar in Fig. 4b), which included global groundwater pumping (the same model configuration as in a previous study[9]) for industrial, irrigation, agricultural, and domestic water demand (Supplementary Fig. 2). We compare this pumping simulation with the first ensemble member of CESM-LE for the same period of 1900–1999 (Fig. 4b, Supplementary Tables 2–3). Since the anthropogenic pumping was not simulated by CESM-LE, the long-term GWS changes can be fully attributed to anthropogenic climate change and internal climate variability.

**Reporting summary**. Further information on research design is available in the Nature Research Reporting Summary linked to this article.

## Data availability
CESM Large Ensemble Community Project generated by NCAR is available at http://www.cesm.ucar.edu/projects/community-projects/LENS/. GRACE/GRACE-FO Mascon data are available at http://grace.jpl.nasa.gov. The GLDAS data used in this study were acquired as part of the mission of NASA's Earth Science Division and archived and distributed by the Goddard Earth Sciences (GES) Data and Information Services Center (DISC).

## Code availability
CESM is a community ESM. The model code for CESM is released http://www.cesm.ucar.edu/models/. Codes for analysis and visualization are available from the authors upon request.

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

## Acknowledgements

This study was supported by MOST 104-2923-M-002-002-MY4 to the National Taiwan University; by the IGEM project "Impact of Groundwater in Earth system models", co-funded by the French Agence Nationale de la Recherche (ANR Grant no. ANR-14-CE01-0018-01) and the Taiwanese Ministry of Science and Technology; by the NASA GRACE Science Team; and by the Research and Technology Development program of the NASA Jet Propulsion Laboratory at the California Institute of Technology, under contract with NASA. M.-H. L. is also supported by MOST 106-2111-M-002-010-MY4.

## Author contributions

M.-H.L. designed the study with contribution from co-authors. W.-Y.W. performed the analysis and drafted the manuscript. W.-Y.W., M.-H.L., Y.W., J.S.F., J.T.R., P.J.-F.Y., A.D., Z.-L.Y. discussed, reviewed, and edited the manuscript.

## Competing interests

The authors declare no competing interests.
