## [Peer Review File · Nature Communications]

Reviewers' comments:

Reviewer #1 (Remarks to the Author):

This study applied the CESM-LE datasets to study the effects of future climate change under the RCP8.5 scenario on the groundwater storage over seven mid-latitude aquifers around the world. The results showed that there are some contrary effects (increased or decreased groundwater storage) occurred in different aquifers, and the effects are not only related to the future precipitation but also related to a lot of hydrological and climatic elements. The value of this study lies on indicating which aquifer may experience stressful condition under the future global warming. It may be possibly used in the management of groundwater resources. I have some suggestions that may improve the manuscript.

General Comments:

1. The main innovation the authors claimed about the study is that they used datasets from fully coupled (land-ocean-atmosphere coupled) simulations rather than using an offline land model simulation. Based on this innovation, I expect some detail comparative analysis between the result from the fully coupled simulation and the result from an offline simulation. Only by this comparison, can we figure out that what we can't get in the offline simulation but could learn from the coupled simulation. Moreover, by some further mechanism analysis about the differences between offline and coupled simulation results, we can check out the exact advantages of using the coupled model and whether it is really necessary to conduct the coupled experiment (which is usually much more expensive than offline experiment) in the research of future groundwater resources.
2. There are 30 members in the CESM-LE ensemble. I think the number of the samples is large enough to do some in-depth uncertainty analysis rather than simply using the standard deviation as the authors did. The authors can show the values of minimum, 25th percentile, 50th percentile, 75th percentile and maximum as well as the degree of agreement among the 30 members throughout the manuscript to help readers better understand the uncertainties and the confidence of their results.

Specific Comments:

L106-108: Why the decrease of snowfall would increase groundwater storage?

L137-138: Please explain why the groundwater recharge increases during the growing season over the Middle East (as shown in Figure S5(i)) since the snowfall declines and the evapotranspiration increases.

L158-167: A key limitation of using climate model in the groundwater research is missing here. Because groundwater flow is driven by the gravity following the topographic ups and downs, accurately reproducing the groundwater table (or storage) pattern needs a high model spatial resolution that can reflect the topographic factors. Most of the climate (land) models cannot run at such high resolution over large-scales. Though some parameterizations are applied in the land models, the reproduced water table pattern is still far away from the results from a 3D-movement groundwater model. Some recent studies have already done some meaningful works to address this issue, such as Fan et al. (2013) in *Science*, Zeng et al. (2016) in *Journal of Advances in Modeling Earth Systems*, De graaf et al. (2017) in *Advances in Water Resources*, just name a few.

L200: Only the emission scenario of RCP8.5 was applied in this study. This is a major limitation of the research. The limitation should be noticed in the main text rather than being referred in the Methods section only.

L217-219: Are the frozen soil, the frequency and intensity of precipitation and land use included in

the CESM-LE? If they are included, maybe showing some preliminary results for these variables over the seven aquifers in the Supplementary Information can give us a first understanding about their relationships with the groundwater storage.

Reviewer #2 (Remarks to the Author):

The paper by Wu et al. investigates groundwater changes in the CESM Large Ensemble (LE) under RCP8.5 relative to present day. They dissect the trends by plotting the individual components of the water balance for several different aquifers across the globe. They conclude that most of the aquifers will see declines from climate change.

While the question asked is an important one, the study here is inadequate on a number of levels and in its current form unacceptable. It leaves the impression of a paper put together hastily and carelessly. I recommend rejection and encourage the authors to substantially revise their paper to demonstrate the robustness of the results before submitting it again.

Individual points:

All the authors do is plot the output from the CESM LE for different basins and describe the plots. They are not evaluating the ability of the model to simulate groundwater, precipitation, evaporation, snow melt, runoff – anything! The paper features no single observational record other than the seasonal cycle of the global total water storage from GRACE (Fig. S1), which is not compared over the same time period with the model and in itself is not a good metric to assess whether a model can be used for future projections. Is the seasonal cycle predictive of the model's ability to simulate a correct response to greenhouse gas forcing? Also, how is global total water storage from a climate model related to how each aquifer is simulated? That seems like a stretch. How many of the other CMIP5 models actually have an aquifer in their groundwater model that deserves the name? Further, according to Fig. S1, CSIRO seems to be the "best model", not CESM, as claimed by the authors. Of course, this is beside the point since the "best model" for total global water storage tells you little about its ability to simulate an individual aquifer, but it illustrates that the authors struggle to interpret the Taylor diagram.

The authors report the groundwater trend value from CESM with a precision of several digits, completely ignoring the huge uncertainties from the groundwater model used, the climate model's uncertain precipitation projection, internal variability, the lack of other climate models, etc. The numbers are not meaningful unless put in context of their uncertainty.

The authors list many possible caveats to their study, like the ones listed above, but that is not enough by far to justify a publication. The authors have to actually investigate whether these caveats affect their conclusions. These results need to be presented in an adequate uncertainty framework, which requires more than simply downloading and plotting the model output. We have learned a lot since the 1990s on different sources of uncertainty in climate model projections, but simply listing them is not enough anymore. Here, it is impossible to say whether the results have any merit and it is upon the authors to demonstrate this.

Is there model drift in groundwater storage? The authors did not even make the effort to look at the CESM preindustrial control simulation to assess this and the role of internal variability absent any climate change.

All supplementary figure should be plotted with the same y-axis, otherwise they are hard to interpret. Preferable would be an actual budget calculation to see what is going on, instead of eye-balling.

The language is incorrect in places and leaves the impression of a lack of experience with climate model data. For example, it is an "ensemble of 30 simulations", not "30 ensembles". Land use change is another factor that matters in future hydroclimate projections, but has been ignored here. What does "contrary effects" in the title mean? The fact that some aquifers have a positive trend and others a negative trend? How is that "contrary"? It is just a different trend for different aquifers. Or was the null hypothesis that all aquifers decline? Why? The authors should define their null hypothesis at the onset of the study.

Finally, the results are not discussed in context of other modeling studies on the topic and what could be done to improve future projections of groundwater.

Reviewer #3 (Remarks to the Author):

Summary

Wu et al. in "Contrary effects of climate change on future groundwater availability in key mid-latitude aquifers" evaluate the CESM-LE set of simulations over the years 2006-2100 under RCP8.5 across seven of the largest mid-latitude groundwater basins that have been deemed to be significantly distressed.

Overall, I think the paper fits within the scope of the Nature Communications and could be, given more work, a valuable contribution to the scientific community. The findings have both scientific and societal impact as groundwater is one of the largest reserves of freshwater globally.

However, I think there are still several major revisions that need to happen prior to this paper being accepted in a top journal such as Nature Communications. I would suggest that the editor assign major revisions to this manuscript.

This decision stems from the fact that the authors:

- 1) Fail to assess the human demand, a critical component. The study needs to try and estimate the anthropogenic component of groundwater extraction and juxtapose that to the climate-driven feedbacks if this study is to have high-impact. This can be done by using Wada et al. [2010] as an example (see Figure 3 and accompanying text in the study).
- 2) Oversell CLM and need to set its performance in the context of all models, not just CMIP5, and discuss its advantages and disadvantages more clearly. Although the authors note they are the first to evaluate an Earth System Model's fidelity in representing groundwater there may precedent as to why (i.e., groundwater modeling community distrusts the simple bucket models in Earth System Models).
- 3) Use changes in snowpack and snowmelt as a key feedback to anchor their evaluation of groundwater recharge across several of the basins. However, snowpack is very poorly resolved in the CESM-LE simulations due to coarse 1-degree grid-cells that don't accurately represent complex terrain. The onus of proof is put on the authors to prove this wrong or eliminate all basins that heavily rely on snowmelt for recharge in this study.
- 4) Need to more comprehensively evaluate the vertical profile of soil moisture, porosity, and infiltration in CLM4, especially under average and extreme precipitation events, across the groundwater basins. A bit of hand-waving is given that increases in precipitation will lead to increased groundwater recharge and sustainability in several of the basins, however, I anticipate the story isn't that simple. Recharge depends on soil porosity and soil moisture which depends on

the vertical distribution of soil type and capillary space which can be influenced by climate feedbacks such as prolonged drought.

Major Critiques

1) I know that projecting anthropogenic influence on groundwater recharge is extremely tough given that policy intervention, technological innovation, and human consumption habits are nonlinear, and in some ways, unpredictable. However, I think there needs to be more of a discussion throughout the main text that juxtaposes the relative influence of climate internal variability and climate change vs the human system in shaping groundwater reserves. This can be done through statistics derived from different studies and by leveraging information from paragraph two of "Validation with temporal variation of groundwater storage" in your supplemental information. In addition, couldn't you use a groundwater depletion scenario approach derived from the "historical anthropogenic groundwater depletion to storage" from Wada et al. [2010] (e.g., Figure 3 from their study) and apply that against the future CESM-LE simulations? You could then build a set of scenarios that emulate historical demand +/- some assumed set of linear/non-linear extraction rates and juxtapose that with the climate-driven impacts. I think anthropogenic influence is a critical missing piece of this study.

2) I agree that a coupled Earth System Model approach to groundwater modeling is advantageous, and really a must, to resolve large-scale teleconnections and important atmosphere-land feedbacks; however, I think the authors oversell CESM as a state-of-the-art groundwater model in lines 69 to 84.

I think lines 69 to 84 need more citations that prove that CESM is:

- a) a state-of-the-art groundwater model
- b) highlight CLM's worth and limitations in the community of models. Worth: it compares well with GRACE against other models with simple-to-intermediate complexity across many hydrologic processes, etc. Limitations: simple bucket model, CLM grid-cells do not communicate to one another, so how can one infer confidence in their ability to represent baseflow across these major groundwater regions?
- c) cite studies that show that CESM is a suitable tool for climate change and hydrologic cycle analysis across the three major feedbacks to groundwater recharge (i.e., snowmelt, evapotranspiration, and precipitation).

3) While I agree that mountain precipitation and snowpack are primary drivers of mid-to-high latitude hydrologic systems and they will likely shift due to climate change, I do not think that these CESM-LENS experiments resolve this adequately for several of the regions, especially the Central Valley. Snowmelt is used throughout the text as a justification of groundwater change, yet these CESM-LE simulations are at a 1-degree resolution and therefore poorly resolve mountainous regions that feed several of the groundwater regions. An argument could be made for region 4 as the Himalayas are sufficiently large enough to be resolved at 1-degree resolution. With that said, if snowmelt is one of the three major feedbacks that replenish groundwater, how can one infer confidence from the simulated long-term trend if a major feedback is poorly resolved (or missing entirely)? To test this, what are average snow depths in the headwaters of these groundwater regions and how do they compare with observed (could be partly sampled via literature review too)? My intuition is that groundwater basins that are heavily reliant on mountain snowmelt recharge cannot be assessed with CESM-LE and should be excluded from this study.

4) It seems that a central argument of the authors is that increased precipitation in the future would lead to more groundwater recharge and potential for sustainability, ignoring human-demand, in groundwater basins such as northern India and China. However, extreme precipitation can exacerbate surface runoff/flooding as it overwhelms capillary retention, especially in years that directly follow drought. This can lead to catastrophic flooding that doesn't beneficially replenish

groundwater. This, drought + extreme precipitation, has happened often in India, a groundwater basin deemed in this study to benefit from increased precipitation. Climate change should only exacerbate this capillary collapse feedback across many of the groundwater basins. Therefore, this study needs a discussion that clarifies the soil characteristics (e.g., porosity) of each groundwater basin assessed within CLM4.

Plot Edits

Figure 1 – This is a very clean and information-rich plot. I think the total GW storage (mm or cm) needs to be shown too. Anomalies are good at showing the trend but can be misleading if they are relatively small compared to total GW storage. Could you place the GW storage plots above/below (i.e., attached along the "time" x-axis) the GW storage anomalies?

Figure 2 – Were the arrow sizes in a) intended to be different sizes to indicate flux magnitude? Could you put some of your Table 1 fluxes in this figure (or derive some from the literature if you deem the simulation to be of poor quality) and size the flux arrows by relative impact? If so, you could either a) lump all basins together and create a total basin flux diagram or b) create a few of these figures with some of the more interesting groundwater basins evaluated or c) lump basins together by climate type or proximity.

Minor Edits

I think the authors fail to convey the worth of CESM-LE in the evaluation of long-term trends to the reader. A description of why this large-ensemble approach is of great value to the scientific community needs to be conveyed. This should be expanded upon, but something like, "CESM-LE effectively samples climate internal variability in CESM through 3,000 years of simulation time which is critical when assessing decadal-to-centennial trends in slow-moving processes such as groundwater storage."

I also think the authors fail to tell readers climate change future they chose to evaluate. In the methods, you state you used RCP8.5, but it needs to be discussed earlier on and could be described more colloquially as a "business-as-usual" scenario.

Line 28-29 – awkward sentence, maybe change to, "Climate change impacts on groundwater storage have the potential to compromise future water availability and sustainability."

Line 30 – "...we used a fully-coupled climate model..."

Line 31 – "...critical aquifers that have been identified as significantly distressed."

Line 41 – "...our results also suggest that there may be potential for enhanced groundwater use in some of the most overstressed aquifers."

Line 52 – "...In terms of the hydrological cycle, climate change can affect the amount of soil infiltration due to changes in extreme precipitation, especially deeper percolation that recharge groundwater."

Line 55 – "...In addition, rising temperatures increase evaporative demand over land which limits the amount of water that can replenish groundwater."

Line 73 – "...provides the simulations used for this study that can be validated against the Gravity Recovery and Climate Experiment (GRACE) observations and other models."

Line 83 – "..., and only account for the natural processes that act on the water balance of the aquifer."

Line 84 – remove "...and define the available supply of water."

In addition, Line 75 is an example of the overstatement of CESM's capabilities I mentioned before. How can CESM be more realistic than GRACE, a satellite derived product that is now a standard in the field?

Line 91 to 93 – Although I agree that climate change will transition precipitation from snow-to-rain and deplete snowpack throughout much of the world's mountainous regions, I don't perceive these simulations to be credible for the assessment of snowmelt in mountainous regions.

Line 106 to 109 – Again, the Sierra Nevada do not resemble a real mountain range at 1-degree grid-resolution, nor do other mountain regions. Studies have shown that the maximum elevation of the Sierra Nevada at 1-degree grid-resolution is ~1600 meters which is almost 3000 meters below the actual maximum. Not only are the Sierra Nevada not resolved, but the California Central Valley is non-existent and artificially uplifted due to the Sierra Nevada being averaged into it at 1-degree resolution.

Line 140 – remove "...(evaporation and transpiration)."

Line 148-150 – awkward set of sentences.

Line 152 – Is this really the first-time groundwater storage has been assessed from a fully-coupled climate model? Why would that be the case given that you intercompared 25 different model simulation(s) in your Taylor Diagram in Supplemental Figure 1?

Line 154 – "While our explanations of key mechanisms followed a more simplistic one-way approach..."

Line 166 – "...complex interactions within the climate system..."

Line 169 – "...changes in precipitation but also other processes modulate the overall evolution (e.g., evapotranspiration, snowmelt, and plant growth).

Line 170 – "Yet, future projections in precipitation are notoriously uncertain, especially in transition zones, where the "rich-get-richer and poor-get-poorer" mechanism does hold, and where groundwater abstraction is structurally important."

Line 176 – "However, it is noteworthy..."

Line 177 – "For instance, changes in timing and/or magnitude of spring snowmelt have been reported in observational data and likely influences groundwater recharge."

Line 181 – northwestern India is topographically complex, but how well is it resolved in CESM-LE?

Replies to reviewer's comments for manuscript NCOMMS-18-07827-T
Divergent effects of climate change on future groundwater availability in key mid-latitude aquifers

Reviewer1

Reviewer's original comments are in black.

The responses are in blue.

Dear Reviewer,

Thanks very much for your advice on manuscript NCOMMS-18-07827-T submitted to Nature Communications for publication. The manuscript has been through significant positive modifications based on your instructions as well as our continuous efforts to make this work better. It is no doubt at all that this work has benefited significantly from your comments, and we believe the revised manuscript is in much better shape than the last one. We sincerely ask you kindly to give us your evaluation and instruction again. In the following please find our detailed response to your comments and the revised manuscript. Your further instructions are much welcomed.

This study applied the CESM-LE datasets to study the effects of future climate change under the RCP8.5 scenario on the groundwater storage over seven mid-latitude aquifers around the world. The results showed that there are some contrary effects (increased or decreased groundwater storage) occurred in different aquifers, and the effects are not only related to the future precipitation but also related to a lot of hydrological and climatic elements. The value of this study lies on indicating which aquifer may experience stressful condition under the future global warming. It may be possibly used in the management of groundwater resources. I have some suggestions that may improve the manuscript.

General Comments:

1. The main innovation the authors claimed about the study is that they used datasets from fully coupled (land-ocean-atmosphere coupled) simulations rather than using an offline land model simulation. Based on this innovation, I expect some detail comparative analysis between the result from the fully coupled simulation and the result from an offline simulation. Only by this comparison, can we figure out that what we can't get in the offline simulation but could learn from the coupled simulation. Moreover, by some further mechanism analysis about the differences between offline and coupled simulation results, we can check out the exact advantages of using the coupled model and whether it is really necessary to conduct the coupled experiment (which is usually much more expensive than offline experiment) in the research of future groundwater resources.

Previous studies have shown the impacts of groundwater on the atmosphere with sensitivity tests, which reveals the advantages of considering the groundwater dynamics in the coupled climate models¹⁻⁶. Lo and Famiglietti (2011)⁵ and Wang et al. (2018)⁶ showed the impacts of groundwater on precipitation patterns with the CESM and IPSL model, respectively^{5,6}. On the other hand, studies have shown the importance of the initial states of land hydrology to climate forecast⁷. Compared to snow or soil moisture, groundwater has much longer residence time and might impact long-term climate projection, which is what we think a fully coupled model (with groundwater component incorporated) should be used when studying the century-long groundwater storage changes.

The above discussion for the comparison of offline and online coupled simulations is included in the discussion of the revised manuscript in *Line 60-70*.

It is challenging to design offline/coupled simulation to detangle the differences based on our framework because the atmospheric fields (forcings) were influenced by groundwater dynamics. CESM with enabled a groundwater model can directly simulate the groundwater changes, while most GCMs cannot. Therefore, we do not need to use the outputs from the CESM (with the groundwater components) to drive the offline simulations again. Our results are based on coupled simulations with land-atmosphere-ocean interactions and feedbacks. This approach is more realistic than a chain of offline simulations. On the other hand, offline simulation is still useful for regional studies with downscaling GCM to a fine spatiotemporal scale.

2. There are 30 members in the CESM-LE ensemble. I think the number of the samples is large enough to do some in-depth uncertainty analysis rather than simply using the standard deviation as the authors did. The authors can show the values of minimum, 25th percentile, 50th percentile, 75th percentile and maximum as well as the degree of agreement among the 30 members throughout the manuscript to help readers better understand the uncertainties and the confidence of their results.

Thanks for the suggestions. We have added the analysis of percentile (the boxplot) based on these 30 members (Fig. R1).

Results show that the amount of groundwater storage changes varies among ensemble members. The ensemble consensus on the sign of changes (+ or -) is all more than 67 percent, indicating the robustness of the ensemble mean. For example, all simulations agree on declines in groundwater over Southern Plains and the Middle East. 67 percent (20 members) agree on increases in groundwater over Central Valley. 83 percent (25 members) agree on increases in groundwater over northwestern India. 97 percent (29 members) agree on increases in groundwater over the North China Plain. All simulations agree on increases over Guarani. 87 percent (26 members) agree on increases in groundwater over Canning Basin.

The above discussion and Figure R1 are included in the revised *Supplementary Information*.

Figure R1. Statistics for 30 ensemble members (min, 25th percentile, median, 75th percentile, max) in each basin. Changes in groundwater in each ensemble member between 2071–2100 and 2006–2035.

Specific Comments:

L106-108: Why the decrease of snowfall would increase groundwater storage?

The original statement was:

In particular, an increase in rainfall and a decrease in snowfall both increase groundwater storage in winter—the season in which groundwater is replenished.

Our original statements were confusing because we meant the “partitioning to rainfall and snowfall”, not the separated effects of rainfall and snowfall.

1. When only decreases of snowfall considered:
Decreases in snowfall lead to declines in snowpack and the amount of snowmelt. This results in less water infiltrate into the soil and less groundwater recharge during/after snowmelt season.
2. When the partitioning “partitioning to rainfall and snowfall” is considered:
This enhances infiltration during snow season.

We have included the above discussions in the revised manuscript to make it clear. Please see the revised manuscript in *Line 116*:

“First, less snowfall with more rainfall (precipitation partitioning) increases GWS in winter (Supplementary Fig. S3); however, the decreased snowmelt reduces groundwater recharge in early spring.”

L137-138: Please explain why the groundwater recharge increases during the growing season over the Middle East (as shown in Figure S5(i)) since the snowfall declines and the evapotranspiration increases.

During JFMA, groundwater recharge is positive (Fig. R2; water moves downward from soil to groundwater when there is infiltration from snowmelt and rain) but getting less in the future mainly due to declines in snowmelt. After May, there is no more replenishment from snowmelt, groundwater recharge is negative (water transports upward from groundwater to soil) and becomes less negative in future (increase as the reviewer mentioned).

Groundwater recharge is a result of gravitational drainage and capillary forces (related to matric potential, water table depth). The reasons for less upward flows might be 1) changes in the redistribution of soil water and groundwater, 2) less root water uptake.

Figure R2. Groundwater recharged over the Middle East.

L158-167: A key limitation of using climate model in the groundwater research is missing here. Because groundwater flow is driven by the gravity following the topographic ups and downs, accurately reproducing the groundwater table (or storage) pattern needs a high model spatial resolution that can reflect the topographic factors. Most of the climate (land) models cannot run at such high resolution over large-scales. Though some parameterizations are applied in the land models, the reproduced water table pattern is still far away from the results from a 3D-movement groundwater model. Some recent studies have already done some meaningful works to address this issue, such as Fan et al. (2013) in Science, Zeng et al. (2016) in Journal of Advances in Modeling Earth Systems, De graaf et al. (2017) in Advances in Water Resources, just name a few.

Thanks for your comments. The discussion of lateral flow has been added in the revised manuscript in Line220-222. Studies have shown groundwater model is required to represent the hillslope hydrology, especially for the high-resolution model (0.1°), which, however, has lateral

very small impacts on water budgets in current GCM resolution (1°)⁸. This justifies our assumption of the impacts of lateral groundwater flow are small given the large horizontal space and the associated small gradient. The impact of lateral flows on the global climate (role in the earth system) still needs more future study as you nicely listed several excellent papers in this regard.

As the reviewer mentioned, a topographic index is one of the parameters in the groundwater model of CLM^{4,5}. Figure R3 shows that the spatial pattern of the water table depth in CESM. Compared to other studies^{8,9}, the spatial patterns are similar but results from CESM is more dominant by climate regime rather than topography,

Figure R3. Water table depth(m) derived from the long-term mean of CESM-LE historical run for 1920–2005.

L200: Only the emission scenario of RCP8.5 was applied in this study. This is a major limitation of the research. The limitation should be noticed in the main text rather than being referred in the Methods section only.

This limitation is discussed in the revised manuscript in *Line 237-241*. NCAR provides a 30-member RCP8.5 Large Ensemble (CESM-LE). There is an RCP4.5 Medium Ensemble (CESM-ME) but with 15 members and only until 2080; thus, we have only analyzed the RCP8.5 simulations.

L217-219: Are the frozen soil, the frequency and intensity of precipitation and land use included in the CESM-LE? If they are included, maybe showing some preliminary results for these variables over the seven aquifers in the Supplementary Information can give us a first understanding about their relationships with the groundwater storage.

Yes, frozen soil and frequency and intensity precipitation are both simulated in CESM-LE simulations. Here we show Southern Plains as an example in Fig. R4. Decreases in surface frozen soil reduce the space of impermeable texture in the soil column and make water easier to

flow through. The thermal and hydrologic properties of soil are also a function of soil moisture, soil texture, and soil temperature. The seven basins in this study are located in middle-latitude regions, where root-zone soil moisture is mostly in liquid water.

Under climate change, extreme precipitation is expected. We analyze the probability distribution of precipitation in the wettest month among 30 years in 30 simulations. At the end of the century, the probability distributions shift to a larger average or the standard deviation increases. We already include impacts of average precipitation to groundwater in the main text. However, to examine how extreme precipitation can affect groundwater storage through changes in soil macropores and fractures needs further studies.

A transient land cover is an input data of CESM, which is prescribed. Therefore, CESM simulates the interactions between precipitation intensity and the LULCC (land use land cover changes) in a one-way approach. Indeed, changes in land cover can change the terrestrial water (urbanization, deforestation). Further sensitivity tests are required to isolate the impacts of different land cover to groundwater.

The discussions mentioned above were added in the *Supplementary Information*.

Figure R4. (a)-(d) Seasonal cycle of snow water equivalent/total soil ice/ total soil liquid water over Southern Plains. Black and red lines represent the ensemble mean/individual ensemble members during 1986–2005 and 2081–2100 (e) Probability distribution of maximum monthly precipitation (f) Time series of transient land cover types in CESM.

Reference:

- 1 Anyah, R. O., Weaver, C. P., Miguez-Macho, G., Fan, Y. & Robock, A. Incorporating water table dynamics in climate modeling: 3. Simulated groundwater influence on coupled land-atmosphere variability. *Journal of Geophysical Research: Atmospheres* **113**, doi:10.1029/2007jd009087 (2008).
- 2 Jiang, X., Niu, G.-Y. & Yang, Z.-L. Impacts of vegetation and groundwater dynamics on warm season precipitation over the Central United States. *Journal of Geophysical Research: Atmospheres* **114**, doi:10.1029/2008jd010756 (2009).
- 3 Maxwell, R. M. *et al.* Development of a Coupled Groundwater–Atmosphere Model. *Monthly Weather Review* **139**, 96-116, doi:10.1175/2010mwr3392.1 (2011).
- 4 Yuan, X., Xie, Z., Zheng, J., Tian, X. & Yang, Z. Effects of water table dynamics on regional climate: A case study over east Asian monsoon area. *Journal of Geophysical Research: Atmospheres* **113**, doi:10.1029/2008jd010180 (2008).
- 5 Lo, M.-H. & Famiglietti, J. S. Precipitation response to land subsurface hydrologic processes in atmospheric general circulation model simulations. *Journal of Geophysical Research: Atmospheres* **116**, doi:10.1029/2010JD015134 (2011).
- 6 Wang, F., Ducharne, A., Cheruy, F., Lo, M.-H. & Grandpeix, J.-Y. Impact of a shallow groundwater table on the global water cycle in the IPSL land–atmosphere coupled model. *Climate Dynamics* **50**, 3505-3522, doi:10.1007/s00382-017-3820-9 (2018).
- 7 Koster, R. D. *et al.* Regions of strong coupling between soil moisture and precipitation. *Science* **305**, 1138-1140 (2004).
- 8 Krakauer, N. Y., Li, H. & Fan, Y. Groundwater flow across spatial scales: importance for climate modeling. *Environmental Research Letters* **9**, 034003, doi:10.1088/1748-9326/9/3/034003 (2014).
- 9 Fan, Y., Li, H. & Miguez-Macho, G. Global Patterns of Groundwater Table Depth. *Science* **339**, 940-943, doi:10.1126/science.1229881 (2013).

Replies to reviewer's comments for manuscript NCOMMS-18-07827-T
Divergent effects of climate change on future groundwater availability in key mid-latitude aquifers

Reviewer #2

Dear Reviewer,

Thanks very much for your advice on manuscript NCOMMS-18-07827-T submitted to Nature Communications for publication. The manuscript has been through significant positive modifications based on your instructions as well as our continuous efforts to make this work better. It is no doubt at all that this work has benefited significantly from your comments, and we believe the revised manuscript is in much better shape than the original version. We sincerely invite you to read our revised manuscript again and kindly provide us your valuable instructions. In the following, please find our detailed response to your comments and the revised manuscript.

Reviewer's original comments are in black.

The responses are in blue.

The paper by Wu et al. investigates groundwater changes in the CESM Large Ensemble (LE) under RCP8.5 relative to present day. They dissect the trends by plotting the individual components of the water balance for several different aquifers across the globe. They conclude that most of the aquifers will see declines from climate change.

While the question asked is an important one, the study here is inadequate on a number of levels and in its current form unacceptable. It leaves the impression of a paper put together hastily and carelessly. I recommend rejection and encourage the authors to substantially revise their paper to demonstrate the robustness of the results before submitting it again.

Individual points:

All the authors do is plot the output from the CESM LE for different basins and describe the plots. They are not evaluating the ability of the model to simulate groundwater, precipitation, evaporation, snow melt, runoff – anything! The paper features no single observational record other than the seasonal cycle of the global total water storage from GRACE (Fig. S1), which is not compared over the same time period with the model and in itself is not a good metric to assess whether a model can be used for future projections. Is the seasonal cycle predictive of the model's ability to simulate a correct response to greenhouse gas forcing? Also, how is global total water storage from a climate model related to how each aquifer is simulated? That seems like a stretch. How many of the other CMIP5 models actually have an aquifer in their groundwater model that deserves the name? Further, according to Fig. S1, CSIRO seems to be the "best model", not CESM, as claimed by the authors.

Of course, this is beside the point since the “best model” for total global water storage tells you little about its ability to simulate an individual aquifer, but it illustrates that the authors struggle to interpret the Taylor diagram.

The main focus of this paper is to examine climate factors that change groundwater variation under warming, not to evaluate the ability of GCMs simulating groundwater variations. CESM is a community model that has been evaluated by many researchers. Kay, et al. ¹ show a few of them (e.g., global surface temperature responses to greenhouse gas is with the spread of simulations and fits well with ensemble mean). We have added some references for developments and evaluations of this model in *Line 71-79 and Methods*.

It is challenging to evaluate large-scale groundwater storage with the in-situ groundwater level because they are conceptually different. We think the best way to compare this 1-degree simulation is using GRACE that captures large-scale dynamics of water storage. Most of the GCMs in CMIP5 did not include groundwater explicitly. CESM groups and GFDL-CM3 are the models representing groundwater explicitly based on our knowledge. Also, “groundwater” is not a standard output in CMIP5 achieve. Therefore, only CESM-LE in Fig. S1 contains groundwater storage.

GRACE data started in 2003; however, the CMIP5 historical run was only until 2005. Due to the constraints, we compared the “long-term climatology” for GRACE and CMIP5, which was seasonal climatology (12-month) of CMIP5 for 25 years and GRACE for 10 years. As shown in *Table R1*, CSIRO-Mk3-6-0 is the model with the smallest RMSE (no explicitly groundwater, 6 soil layers, to 4.6m deep), and the CESM-LE (explicitly groundwater, 10 hydrologically-active soil layers to 3.8m) is the one with second smallest. In the revised SI, we clarified CESM as one of the “realistic” models.

Table R1. The correlation coefficient (r), normalized standard deviation (ratio of the model to observation), root mean square error in Fig. S1

	r	Normalized STD (ratio)	RMSE
CSIRO-Mk3-6-0	0.923	0.948	0.637
CESM-LE	0.923	1.278	0.860
CanESM2	0.856	0.980	0.875
MIROC5	0.908	1.279	0.921
GFDL-CM3	0.908	1.337	0.986
CNRM-CM5	0.875	1.245	1.004
CESM1-CAM5	0.883	1.278	1.012
inmcm4	0.840	1.198	1.069
MIROC4h	0.862	1.273	1.076

MPI-ESM-MR	0.824	1.195	1.115
MPI-ESM-LR	0.823	1.244	1.164
CESM1-WACCM	0.805	1.196	1.169
CCSM4	0.850	1.395	1.248
CESM1-BGC	0.857	1.414	1.251
CESM1-FASTCHEM	0.842	1.396	1.274
NorESM1-M	0.806	1.370	1.345
MIROC-ESM-CHEM	0.822	1.488	1.443
MIROC-ESM	0.801	1.481	1.493
BNU-ESM	0.698	1.261	1.500
MRI-CGCM3	0.817	1.533	1.513
FGOALS-g2	0.634	1.292	1.672
bcc-csm1-1-m	-0.962	0.492	2.436
bcc-csm1-1	-0.965	0.761	2.874
GISS-E2-R	0.884	2.619	2.960

The authors report the groundwater trend value from CESM with a precision of several digits, completely ignoring the huge uncertainties from the groundwater model used, the climate model's uncertain precipitation projection, internal variability, the lack of other climate models, etc. The numbers are not meaningful unless put in context of their uncertainty.

We appreciate the comments. We have addressed the uncertainty in different approaches:

- (1) In *Fig. 2* and *Fig. S3-S9*, the shaded areas represent a single standard deviation among the ensemble of 30 simulations and Table. S2 has been added to represent the uncertainties of Table1. The uncertainties of groundwater trends have been quantified and shown in the revised manuscript.
- (2) We have added the analysis of probability based on these 30 ensemble members (Figure. R1) in *Supplementary Information*. Results show that most of the simulations agree on the same sign of changes (The ensemble consensus on the sign of changes is more than 66%); however, the amount of groundwater storage changes varies among the ensemble. All simulations agree on declines in groundwater over Southern Plains and the Middle East. 67% (20 members) agree on increases in groundwater over Central Valley. 83% (25 members) agree on increases in groundwater over northwestern India. 97% (29 members) agree on increases in groundwater over the North China Plain. All simulations agree on increases over Guarani. 87% (26 members) agree on increases in groundwater over Canning Basin.

(3) We revised the manuscript with more focus on mechanisms driven groundwater changes rather than quantification of groundwater changes.

The above uncertainty analyses have been incorporated into the revised manuscript.

Figure R1. Statistics for 30 ensemble members (min, 25th percentile, median, 75th percentile, max) in each basin. Changes in groundwater in each ensemble member between 2071–2100 and 2006–2035.

The authors list many possible caveats to their study, like the ones listed above, but that is not enough by far to justify a publication. The authors have to actually investigate whether these caveats affect their conclusions. These results need to be presented in an adequate uncertainty framework, which requires more than simply downloading and plotting the model output. We have learned a lot since the 1990s on different sources of uncertainty in climate model projections, but simply listing them is not enough anymore. Here, it is impossible to say whether the results have any merit and it is upon the authors to demonstrate this.

Is there model drift in groundwater storage? The authors did not even make the effort to look at the CESM preindustrial control simulation to assess this and the role of internal variability absent any climate change.

The CESM-LE simulation after 1850 was released, and there was a 400-yr simulation before 1850. In this study, we only take data after 1900. Our analysis (*Fig. R2*) shows that the trend is induced by warming, not by the spin-up process, or climate drift. We compared a simulation of pre-industrial with another simulation of historical and RCP8.5 for 250 years. Two simulations are started with the same land initial value from a previous pre-industrial simulation (see *Table 1* in Kay, et al. ¹). Within the same length of the period, the evolutions of groundwater storage are significantly different. This indicates the responses are from greenhouse gas effects, and the model drifting could be neglected. Also, the time series from the pre-industrial simulation is stable and has no apparent trend compared to one from historical+RCP8.5, which we use in this study.

Fig. R2 250-yr annual-average time series of groundwater storage for pre-industrial (black) and for historical+RCP8.5 (red) over Southern Plains

Table R2 shows another statistical approach² to check the spin-up of the groundwater. Results show that the year-to-year groundwater variation is smaller than 0.1%–10% of the annual average groundwater storage. The variation mainly comes from natural inter-annual variability, and the climate drift in groundwater is negligible.

Table R2 Checking the spin-up of groundwater storage, Z is the annual mean of groundwater storage over the studied regions

Preindustrial simulation	$\frac{ Z_{1851} - Z_{1850} }{Z_{1850}}$
1. Central Valley	$<10^{-3}$
2. Southern Plains	$<10^{-2}$
3. Middle East	$<10^{-3}$
4. Northwestern India	$<10^{-2}$

5. North China Plain	$<10^{-3}$
6. Guarani	$<10^{-2}$
7. Canning	$<10^{-1}$

All supplementary figure should be plotted with the same y-axis, otherwise they are hard to interpret. Preferable would be an actual budget calculation to see what is going on, instead of eye-balling.

Thanks for the comments. However, the values for those variables differ tremendously among 7 aquifers. The groundwater budget includes the input of the recharge minus output of baseflow. Alternatively, we quantify the impact of different climate factors on groundwater by summarizing the climate-driven groundwater storage trends that could be related to precipitation, evapotranspiration, and snowmelt. Therefore, *Fig. 3 (Fig. R3 below)* was added in the revised manuscript with a water-budget-based regression to quantify different component's impacts on groundwater recharge. Overall, changes in groundwater recharge are dominant by rainfall over monsoon and humid regions. The distribution of regions dominated by snow depends on latitude and elevation. Over dry regions, changes in evapotranspiration are the dominant factors to groundwater recharge.

Fig R3. The relative contribution of three climate-driven factors (rainfall, snowmelt, and evapotranspiration (ET)) to groundwater recharge derived from the regressions based on the CESM-LE simulation data. The projected future trends (2006-2010) under the business-as-usual scenario (RCP8.5). Each point represents three coefficients based on snowmelt (red), ET (green),

and rainfall (blue). Region-averaged results are labeled on the color bar with the number of locations shown in Fig. 2. For example, the results of Guarani (6) is (snowmelt: 0.0, ET: 0.4, rainfall:0.6).

Method:

The impacts of three climate-driven factors (rainfall, snowmelt, ET) on groundwater recharge is calculated via the multiple linear regression:

$$y = a_1x_1 + a_2x_2 + a_3x_3 + b \quad (1)$$

where y is the standardized annual groundwater recharge, and x_i is the corresponding standardized annual rainfall, snowmelt, and ET at each global grid after applying the 11-year running mean. Only the regions with the statistical significance at a 90% confidence level are shown in Fig. 3. The contributions of three climate-driven factors to groundwater recharge is quantified by the following coefficient ranging between 0 and 1:

$$Contribution_i = \frac{|a_i|}{\sum_{i=1}^3 |a_i|} \quad (2)$$

The results are plotted as RGB (R: snowmelt, G: ET, B: rainfall) for triplet color in Fig. 5 (also see Supplementary Fig. S11). For example, the aquifer-averaged results of Canning Basin (7) are (0, 0.36, 0.64), and the results of Southern Plains (2) is (0.30, 0.24, 0.46).

The language is incorrect in places and leaves the impression of a lack of experience with climate model data. For example, it is an “ensemble of 30 simulations”, not “30 ensembles”. Land use change is another factor that matters in future hydroclimate projections, but has been ignored here. What does “contrary effects” in the title mean? The fact that some aquifers have a positive trend and others a negative trend? How is that “contrary”? It is just a different trend for different aquifers. Or was the null hypothesis that all aquifers decline? Why? The authors should define their null hypothesis at the onset of the study.

We corrected throughout the text and supplementary. We thank the reviewer for his/her comments.

We have discussed the land-use change in *Supplementary Information*. CESM-LE includes transient land cover changes (Fig. S12-19). The title of “Contrary” means under global warming, both negative and positive trends of groundwater storage can happen with increases in rainfall. We found that changes in groundwater storage do not only be contributed by changes in precipitation but also by other factors (evapotranspiration, snowmelt). The title of this paper has been revised to “*Divergent effects of climate change on future groundwater availability in key mid-latitude aquifers*”.

Finally, the results are not discussed in context of other modeling studies on the topic and what could be done to improve future projections of groundwater.

We incorporate several suggestions for improvements in the modeling in the discussion section in *Line 220-241* in the revised manuscript about:

- (1) Representation of lateral flow exchanges between grid cells in the land surface models.
- (2) Higher resolution or sub-grid approach for representation complex terrain through the downscaling approaches.
- (3) Scenario developing for anthropogenic pumping
- (4) Investigation of other factors such as land cover changes

Reference:

- 1 Kay, J. E. *et al.* The Community Earth System Model (CESM) Large Ensemble Project: A Community Resource for Studying Climate Change in the Presence of Internal Climate Variability. *Bulletin of the American Meteorological Society* **96**, 1333-1349, doi:10.1175/bams-d-13-00255.1 (2015).
- 2 Yang, Z. L., Dickinson, R., Henderson-Sellers, A. & Pitman, A. Preliminary study of spin-up processes in land surface models with the first stage data of Project for Intercomparison of Land Surface Parameterization Schemes Phase 1 (a). *Journal of Geophysical Research: Atmospheres* **100**, 16553-16578 (1995).

Replies to Reviewer's comments for manuscript NCOMMS-18-07827-T
Divergent effects of climate change on future groundwater availability in key mid-latitude aquifers

Reviewer #3

Reviewer's original comments are in black.

The responses are in blue.

Dear Reviewer,

Thanks very much for your advice on manuscript NCOMMS-18-07827-T submitted to Nature Communications for publication. The manuscript has been through significant positive modifications based on your instructions, as well as our continuous efforts to make this work better. It is no doubt at all that this work has benefited significantly from your comments, and we believe the revised manuscript is in much better shape than the original version. We sincerely invite you to read our revised manuscript again and kindly provide us your valuable instructions. In the following, please find our detailed response to your comments and the revised manuscript.

Summary

Wu et al. in "Contrary effects of climate change on future groundwater availability in key mid-latitude aquifers" evaluate the CESM-LE set of simulations over the years 2006-2100 under RCP8.5 across seven of the largest mid-latitude groundwater basins that have been deemed to be significantly distressed.

Overall, I think the paper fits within the scope of the Nature Communications and could be, given more work, a valuable contribution to the scientific community. The findings have both scientific and societal impact as groundwater is one of the largest reserves of freshwater globally.

However, I think there are still several major revisions that need to happen prior to this paper being accepted in a top journal such as Nature Communications. I would suggest that the editor assign major revisions to this manuscript.

This decision stems from the fact that the authors:

1) Fail to assess the human demand, a critical component. The study needs to try and estimate the anthropogenic component of groundwater extraction and juxtapose that to the climate-driven feedbacks if this study is to have high-impact. This can be done by using Wada et al. [2010] as an example (see Figure 3 and accompanying text in the study).

Thank you for the suggestion. As suggested, we added the assessment of anthropogenic pumping + climate driven in the revised manuscript (Fig. R1, see below). Orange bars in Fig. R1b utilized the CESM groundwater abstraction simulation from Wada, et al. ¹. The estimated groundwater abstraction data from Wada, et al. ² has been applied to the same coupled modeling framework.

Fig. R1 (Fig. 4 in the revised manuscript) *Trends of GWS.* (a) GRACE-based estimates (i.e., GRACE minus GLDAS) of GWS trends (2003–2014). (b) The 20th-century GWS trends with pumping are estimated based on CESM simulations (1900–1999) in our previous study⁴⁰. The 20th and 21st-century GWS trends without pumping are estimated based on the CESM-LE simulation during 1900–1999 and during 2006–2100, respectively. The error bars represent the spread of one standard deviation among 30 ensemble members.

Here we compare the results from three sets of simulations to separate “climate-driven” (orange) and “climate-driven + anthropogenic pumping” (blue) in the 20th century, and a 21st-century

projection due to climate-driven effects without anthropogenic pumping (green). Generally, pumping leads to negative groundwater trends, especially over the time and place that pumping happened (the Central Valley, the Southern Plains, the Middle East, Northwestern India, and the North China Plain). In non-pumping regions, changes are associated with atmospheric circulations changes in the coupled earth system model (the Guarani Aquifer and the Canning Basin). Larger negative groundwater trends in a recent decade compared to the entire 20th century is associated with more extensive water usage. By comparing “climate-driven” and “climate-driven + anthropogenic pumping” in the 20th century, we find that the contribution of anthropogenic pumping could easily far exceed the natural replenishment. Therefore, it is essential to understand the mechanisms between climate factors and groundwater and how climate factors control the availability of groundwater under future warming.

We have included the above discussion and Fig. R1 into the revised manuscript.

2) Oversell CLM need to set its performance in the context of all models, not just CMIP5, and discuss its advantages and disadvantages more clearly. Although the authors note they are the first to evaluate an Earth System Model's fidelity in representing groundwater there may precedent as to why (i.e., groundwater modeling community distrusts the simple bucket models in Earth System Models).

We chose CESM as our tool because of (1) To resolve fully interaction in the earth system (2) The flexibility of cross-basin, global study. NCAR CLM group has been working on resolving hillslope hydrology in sub-grid but has not released in newer versions.

We have revised the manuscript and made sure not overselling CESM. Please see the revised manuscript in *Line 60-79 and Discussion*. We have also done a literature review on the comparisons between LSMs and GHMs summarized as below.

The development purpose of LSMs (land surface models) and GHMs (global hydrology models) models are different. LSMs as an integrated part of the Earth System Modelling (ESM) was developed to provide land surface fluxes to the atmospheric models, while GHMs were designed for the estimation of water resources. Due to the purpose of developing GHMs, and most of the previous studies of global water management relied on GHMs. For example, the future projection of groundwater recharge was assessed³ using CMIP5-WaterGAP^{4,5} model, which used the forcing data simulated by different GCMs in CMIP to drive a GHM, the WaterGAP model.

However, some recent studies showed that there are no significant differences between LSMs and GHMs in terms of simulating water fluxes or storages. Water Model Intercomparison Project (WaterMIP)^{4,5} put GHMs and LSMs together for comparison. They found there are no significant differences for evapotranspiration and runoff between the LSMs and GHMs. Also, globally averaged GHMs and global climate models contribute to a similar spread⁵.

A recent study⁶ also shows (1) both GHMs and LSMs have similar performances compared to GRACE across large-river basins (2) after sorting 186 large river basins from decreasing TWS trend to increasing TWS trend based on GRACE, compared to other LSMs, CLM4.0 has the closest agreement with GRACE for the basins with increasing trends⁶. To investigate current large-scale groundwater changes, we can rely on satellites and in-situ observations. However, to distinguish climate-driven effects and human footprint (e.g., pumping), model simulations are required. Furthermore, for the future projections, hydrology/groundwater communities still rely on atmospheric inputs from the projection of earth system modeling such as CMIP5. We believe our single earth system modeling approach is more suitable for interpreting results with physical mechanisms.

In summary, we think groundwater modeling is powerful for regional studies with comprehensive geology information and calibrated by observations from wells and streamflow. However, no significant differences exist between GHMs and LSMs for large-scale studies; thus, we chose a more suitable LSM (CLM in this case since it includes a simple groundwater model component) in a single ESM framework. We have added the above discussion of the advantages and disadvantages of LSMs in ESMs versus offline GHMs in the main text in *Line 60-70 and Line 220-222* in the main text of the revised manuscript.

3) Use changes in snowpack and snowmelt as a key feedback to anchor their evaluation of groundwater recharge across several of the basins. However, snowpack is very poorly resolved in the CESM-LE simulations due to coarse 1-degree grid-cells that don't accurately represent complex terrain. The onus of proof is put on the authors to prove this wrong or eliminate all basins that heavily rely on snowmelt for recharge in this study.

The coarse resolution (0.9x1.25 degree) of CESM might be a problem to represent the spatial heterogeneity of snow over the complex terrain. This is one of the disadvantages of using Earth System Models. Here we focus on a spatial average of more than 206,950 km² (18 grid cells for Central Valley in CESM). We assume this feature of spatial heterogeneity only plays an essential role at the sub-grid scale but can be ignored for our current purpose. Here we show the impacts of resolution on the snow simulation (SWE) with GLDAS-Noah⁷ under 1-degree and 0.25-degree for a similar basin (*Fig. R2*). Results show that both resolutions can show the temporal evolution of SWE (seasonality, interannual variability, and trend). Overall, we think downscaling

for hillslope simulation is possible but not necessary for the estimations of the basins averaged trends, especially for the sign of the trend.

Fig. R2 Snow water equivalent (SWE) over California Watersheds in GLDAS-Noah (top) 0.25-degree resolution (bottom) 1-degree resolution. Data and figures were processed by NASA Giovanni

The snow parameterizations in CLM are based on Anderson ⁸; Jordan ⁹; Dai and Zeng ¹⁰. The major reference for parameterization improvements of CLM4.0 is Lawrence, et al. ¹¹. Groundwater scheme is added based on Niu, et al. ¹². The snow parameterization in CLM4.0 has comprehensive physical processes with mass and energy balance. For example, CLM4.0 simulates ice and water in the snowpack, black carbon, and dust within the snow and meltwater flushing with multiple snow layers. Also, inside a single grid, different plant functional types (PFT tile mosaics) are considered in CLM4.0. Thus, we tend to have all the 7-basin remained in this study, and we have incorporated the above discussion and the uncertainty of simulations in the revised manuscript particular in *Line 220 to 222*.

4) Need to more comprehensively evaluate the vertical profile of soil moisture, porosity, and infiltration in CLM4, especially under average and extreme precipitation events, across the groundwater basins. A bit of hand-waving is given that increases in precipitation will lead to increased groundwater recharge and sustainability in several of the basins, however, I anticipate the story isn't that simple. Recharge depends on soil porosity and soil moisture which depends

on the vertical distribution of soil type and capillary space which can be influenced by climate feedbacks such as prolonged drought.

In all of the studied regions except for Northwestern India (*Table R1*), we see a uniformly (surface and deep) and continually increasing or decreasing trend in all water storages (surface soil moisture, deep soil moisture, and groundwater). Soil moisture profiles (*Fig. R3*) show drying in upper layers but wetting in lower layers over Northwestern India. Over Northwestern India, soil moisture and groundwater (*Fig. 2*) first decrease for several decades and then increase later, and different layers show different turning point (stationary points). Such non-uniformly drying and wetting might also be associated with high spatial heterogeneity over this region– drying in the northern part and wetting in the southern part (*Fig. S2*). We acknowledge that clay-rich vertisols and other climate-induced soil texture changes have been neglected in LSMs. We hope we can do future research on that. In this paper, we focus on the direct climate-driven factors only.

Table R1. Trends of soil moisture and groundwater from CESM-LE for 2006-2100

Variable	Unit	1	2	3	4	5	6	7
		Central Valley	Southern Plains	Middle East	Northwestern India	North China Plain	Guarani	Canning Basin
Soil moisture at 20cm ¹	mm/dec	0.01	-0.11	-0.11	-0.06	0.04	0.17	0.02
Soil moisture at 1m ²	mm/dec	0.05	-1.12	-0.46	-0.01	0.29	1.73	0.12
Groundwater recharge	mm/yr/dec	5.5	-0.2	-0.6	2.0	3.7	0.7	3.3
Groundwater storage	mm/dec	1.8	-23.3	-15.2	7.4	10.0	54.3	3.6

1: soil layer thickness of 12cm; 2: soil layer thickness of 55cm.

Fig. R3 Long-term average soil moisture profile over Northwestern India. Black and red lines represent the ensemble mean during 2006–2035 (more recent) and 2071–2100 (future), respectively. The figure shows a drying above 1.5 m, and wetting below it. The modeled water table depth in the regions is below 5 m.

The impact of the changing climate on soil texture is an interesting and vital point. However, our modeling approach is not suitable enough to represent this geologic impact. CESM-LE includes transient land cover changes but no transient soil. The soil texture may change over time, but it is not included in the model yet. In the model, increases in precipitation extremes could only change soil conductivity via soil moisture but not the partition of sand/clay (texture). We acknowledge the issue raised by the reviewer, and we have included the above discussion in the revised manuscript in *Line 228-232*.

Major Critiques

1) I know that projecting anthropogenic influence on groundwater recharge is extremely tough given that policy intervention, technological innovation, and human consumption habits are nonlinear, and in some ways, unpredictable. However, I think there needs to be more of a discussion throughout the main text that juxtaposes the relative influence of climate internal variability and climate change vs the human system in shaping groundwater reserves. This can be done through statistics derived from different studies and by leveraging information from paragraph two of “Validation with temporal variation of groundwater storage” in your supplemental information. In addition, couldn’t you use a groundwater depletion scenario approach derived from the “historical anthropogenic groundwater depletion to storage” from Wada et al. [2010] (e.g., Figure 3 from their study) and apply that against the future CESM-LE

simulations? You could then build a set of scenarios that emulate historical demand +/- some assumed set of linear/non-linear extraction rates and juxtapose that with the climate-driven impacts. I think anthropogenic influence is a critical missing piece of this study.

We agree that anthropogenic pumping is essential and should be investigated. We added the analysis using data from Wada, et al. ¹ to explore the climate-driven and anthropogenic impacts on groundwater storage (Fig. 4) for the present day. Our results show that in these overstressed regions, current groundwater depletion is mainly caused by anthropogenic pumping. However, it is challenging to include pumping in the future projection. Currently, there is not enough data to project groundwater pumping. Hence, we equate the current and future climate impacts on groundwater to the existing pumping, which suggests a sort of minimum impact expected from pumping for the future period. It is hard to expect pumping will decrease given the increasing population and developing economy. Thus, due to the high uncertainty and in the future projection of those human water management, we only investigate both climate-driven and anthropogenic impacts on groundwater for the 20th century. We have further illustrated the importance of the anthropogenic pumping into the revised main content.

2) I agree that a coupled Earth System Model approach to groundwater modeling is advantageous, and really a must, to resolve large-scale teleconnections and important atmosphere-land feedbacks; however, I think the authors oversell CESM as a state-of-the-art groundwater model in lines 69 to 84.

I think lines 69 to 84 need more citations that prove that CESM is:

- a) a state-of-the-art groundwater model
- b) highlight CLM's worth and limitations in the community of models. Worth: it compares well with GRACE against other models with simple-to-intermediate complexity across many hydrologic processes, etc. Limitations: simple bucket model, CLM grid-cells do not communicate to one another, so how can one infer confidence in their ability to represent baseflow across these major groundwater regions?
- c) cite studies that show that CESM is a suitable tool for climate change and hydrologic cycle analysis across the three major feedbacks to groundwater recharge (i.e., snowmelt, evapotranspiration, and precipitation).

This part has been modified as below (*Line 71-95*) with the highlighted section of the reviewer suggests

In this paper, rather than using an offline model simulation, we use the fully-coupled simulations in the Community Earth System Model- Large Ensemble Project (CESM-LE)¹³. The CESM is a fully-coupled climate model, including the land, atmosphere, ice, and ocean components, designed to simulate climate changes with internal climate

variability¹³. The large ensemble (30 members used in this study) approach accounts for uncertainties within the same climate model. Simulations with the length of thousands of years are critical for assessing the decadal-to-centennial trends of the slow-moving, long-memory processes, such as GWS. CESM-LE has been continuously developed¹¹, evaluated¹⁴, and broadly used for investigating the terrestrial water cycle¹⁵ and its components, such as snowpack^{14,16,17}, soil moisture¹⁸, snowmelt runoff¹⁹, and water availability²⁰.

A physically-based groundwater parameterization is embedded in version 4.0 of the Community Land Model (CLM4.0)^{12,21,22}, which is the land surface model of CESM. CLM4.0 has been developed and evaluated with the Gravity Recovery and Climate Experiment (GRACE) observations¹² (as shown in Supplementary Fig. S1). By simulating the water table depth, groundwater recharge and discharge, and the interactions with the overlying soils, the groundwater parameterization in CLM4.0 can model the physical dynamics of storage changes in the unconfined aquifer¹², which is an essential part of terrestrial water storage^{23,24}. Owing to its coupling with the atmospheric and ocean models, CESM is a suitable tool for resolving the multiple interactions and feedbacks of the groundwater system within the hydrologic cycle under a fully-coupled Earth system framework^{1,25,26}.

The limitation of CESM has been added to the discussion.

(a) The development of CLM (from CLM3. 0²⁷ to CLM4.0^{22,28}) shows significant improvement of modeled TWS^{11,12,28} after including the groundwater component¹².

In the main text, we cited the major reference of CESM-LE:

Kay, J. E. *et al.* The Community Earth System Model (CESM) Large Ensemble Project: A Community Resource for Studying Climate Change in the Presence of Internal Climate Variability. *Bull. Am. Meteorol. Soc.* **96**, 1333–1349 (2015).

Major references for CLM4.0:

Oleson, Keith, et al. "Technical description of the community land model (CLM)." (2004).

Oleson, Keith W., et al. "Technical description of version 4.0 of the Community Land Model (CLM)." (2010).

Oleson, K. *et al.* Improvements to the Community Land Model and their impact on the hydrological cycle. *J. Geophys. Res.* **113**, G01021 (2008).

And the development of groundwater scheme with GRACE:

Niu, G.-Y., Yang, Z.-L., Dickinson, R. E., Gulden, L. E. & Su, H. Development of a simple groundwater model for use in climate models and evaluation with Gravity Recovery and Climate Experiment data. *J. Geophys. Res.* **112**, D07103 (2007).

(b) Based on our knowledge, CLM4.0 is one of the very few LSMs in GCMs that explicitly resolve groundwater storage with open code/data access. The resolution of CESM-LE is 0.9*1.25 both for the atmosphere model, as well as the land model. At this spatial scale, we assume the lateral groundwater flow might not be important as a high-resolution hydrologic model for mapping spatial heterogeneity, which is not our purpose. The limitation of one-dimensional LSM has been added in the discussion as below:

(c) Recent studies of using CESM to assess changes of terrestrial hydrology cycle have been cited as below:

- Zeng, Z., Piao, S., Li, L. Z., Wang, T., Ciais, P., Lian, X., ... & Myneni, R. B. (2018). Impact of Earth greening on the terrestrial water cycle. *Journal of Climate*, 31(7), 2633-2650.
- Fyfe, J. C., Derksen, C., Mudryk, L., Flato, G. M., Santer, B. D., Swart, N. C., ... & Scinocca, J. (2017). Large near-term projected snowpack loss over the western United States. *Nature communications*, 8, 14996.
- Mudryk, L. R., Kushner, P. J., Derksen, C., & Thackeray, C. (2017). Snow cover response to temperature in observational and climate model ensembles. *Geophysical Research Letters*, 44(2), 919-926.
- Cheng, S., Huang, J., Ji, F., & Lin, L. (2017). Uncertainties of soil moisture in historical simulations and future projections. *Journal of Geophysical Research: Atmospheres*, 122(4), 2239-2253.
- Mankin, J. S., Viviroli, D., Singh, D., Hoekstra, A. Y., & Diffenbaugh, N. S. (2015). The potential for snow to supply human water demand in the present and future. *Environmental research letters*, 10(11), 114016.
- Ferguson, C. R., Pan, M., & Oki, T. (2018). The effect of global warming on future water availability: CMIP5 synthesis. *Water Resources Research*, 54(10), 7791-7819.
- Rhoades, A. M., Ullrich, P. A., & Zarzycki, C. M. (2018). Projecting 21st century snowpack trends in western USA mountains using variable-resolution CESM. *Climate Dynamics*, 50(1-2), 261-288.

3) While I agree that mountain precipitation and snowpack are primary drivers of mid-to-high latitude hydrologic systems and they will likely shift due to climate change, I do not think that these CESM-LENS experiments resolve this adequately for several of the regions, especially the Central Valley. Snowmelt is used throughout the text as a justification of groundwater change, yet these CESM-LE simulations are at a 1-degree resolution and therefore poorly resolve mountainous regions that feed several of the groundwater regions. An argument could be made for region 4 as the Himalayas are sufficiently large enough to be resolved at 1-degree resolution. With that said, if snowmelt is one of the three major feedbacks that replenish groundwater, how can one infer confidence from the simulated long-term trend if a major feedback is poorly

resolved (or missing entirely)? To test this, what are average snow depths in the headwaters of these groundwater regions and how do they compare with observed (could be partly sampled via literature review too)? My intuition is that groundwater basins that are heavily reliant on mountain snowmelt recharge cannot be assessed with CESM-LE and should be excluded from this study.

Thanks for your thoughtful suggestion. We have added the discussion of the limitation of coarse resolution. The ability to represent snowmelt, mainly due to the accuracy of topography was cited in 215-222. The snow processes have been sophisticatedly simulated in CLM, including the partitioning of precipitation into rainfall and snowfall, snow-albedo feedback, dust impaction snow, snow aging, snow compaction, snow darkening effect, and partitioning of snowmelt into runoff and infiltration on frozen ground. Compared to other LSMs in advanced Weather Research and Forecasting (WRF), CLM better represents snowmelt because of a more realistic surface energy equation for snowmelt²⁹. We think physical feedback is not poorly resolved. The concern is the degraded elevation of a mountain peak from coarser-resolution might underestimate the importance of snowmelt. Therefore, in the revised manuscript, we emphasize the importance of considering all climate factors in a global view (Fig. R4). And, we indicated that the quantification of the results might be model dependent. In Fig. R4, the contribution of snowmelt (red) is robust in mountainous regions (the Rocky Mountains, southern Andes, Himalayas, and the Alps).

Thanks for the suggestion of the literature review. Overall, models with smoother terrain simulated lower SWE at mountain peak³⁰. However, a study also suggested that the snow cover extent continental the United States has a more substantial bias than the western Siberia¹⁴. This gives us confidence in choosing Central Valley as one of our study regions.

Northwestern India, another studied region that the reviewer concerned, is highly influenced by the India monsoon, which is relied on the simulation of atmospheric circulation from climate models. A recent study also shows the importance of replenishment from the rainfall runoff will moderate the decreased snow resource over the Indus Basin with a similar modeling approach¹⁹.

Fig. R4 The relative contribution of three climate-driven factors (rainfall, snowmelt, and evapotranspiration (ET)) to groundwater recharge derived from the regressions based on the CESM-LE simulation data. The projected future trends (2006-2010) under the business-as-usual scenario (RCP8.5). Each point represents three coefficients based on snowmelt (red), ET (green), and rainfall (blue). Region-averaged results are labeled on the color bar with the number of locations shown in Fig. 2. For example, the results of Guarani (6) is (snowmelt: 0.0, ET: 0.4, rainfall:0.6). (Fig. 3 in the revised manuscript)

4) It seems that a central argument of the authors is that increased precipitation in the future would lead to more groundwater recharge and potential for sustainability, ignoring human-demand, in groundwater basins such as northern India and China. However, extreme precipitation can exacerbate surface runoff/flooding as it overwhelms capillary retention, especially in years that directly follow drought. This can lead to catastrophic flooding that doesn't beneficially replenish groundwater. This, drought + extreme precipitation, has happened often in India, a groundwater basin deemed in this study to benefit from increased precipitation. Climate change should only exacerbate this capillary collapse feedback across many of the groundwater basins. Therefore, this study needs a discussion that clarifies the soil characteristics (e.g., porosity) of each groundwater basin assessed within CLM4.

We think extreme precipitation might affect the partitioning of precipitation to infiltration but not to the degree of affecting the signs of trends. It happens in some events in the rainy season, which reveals in the sub-daily-to-seasonal scale but not in the long-term trend of annual mean that we focused on the main text. We agree that changes in the condition of the soil (water content, temperature, and texture) might affect the infiltration rate¹¹, such as higher flooding risk after the drought. However, this is not significantly revealed in our analysis of the 95-year trend and basin average.

To clarify, we compute the year-to-year runoff ratio (R/P) in the studied basins. Overall, we see decreases in runoff ratio in arid basins (Southern Plains, and the Middle East) and increases in runoff ratio in humid regions under warming (Fig. R6, for example)³¹. Over Northwestern India and North China Plains, we see increases in runoff and runoff ratio but also increases in infiltration and groundwater recharge. Even though there are small increases in runoff ratio and more infiltration, and as the results, more water stores in groundwater aquifer. We considered the importance of increasing atmospheric water demand (ET, which is a larger amount compared to runoff). We acknowledge the issue raised by the reviewer, and we have included the above discussion in the revised manuscript in Supplementary Information.

Fig. R6 15-yr running mean of precipitation (mm/yr) and runoff ratio (total runoff/total precipitation) over Northwestern India and North China Plain.

Plot Edits

Figure 1 – This is a very clean and information-rich plot. I think the total GW storage (mm or cm) needs to be shown too. Anomalies are good at showing the trend but can be misleading if they are relatively small compared to total GW storage. Could you place the GW storage plots above/below (i.e., attached along the "time" x-axis) the GW storage anomalies?

It is unknown how much groundwater globally, and the groundwater component in the CLM of CESM won't be able to reveal such information. To assess the uncertainties of internal climate variability in future projections, we further analyzed the uncertainties through the spreads among the simulations of 30 ensemble members in CESM-LE. Results show that the ranges of simulated groundwater storage changes among ensemble members vary by different aquifers. (Fig. R7). The ensemble consensus on the sign of changes (+ or -) is all larger than 67% for all seven aquifers considered, indicating the robustness of the ensemble averages. For example, all 30 members simulated the declines (increases) in groundwater storage over Southern Plains and the Middle East (Guarani). In addition, 20, 25, 28 and 29 members simulated increasing groundwater storage over Central Valley, northwestern India, Canning Basin, and North China Plain, respectively. Above discussion has been added to *Supplementary Information*.

Fig. R7 Differences in groundwater storage during 2071–2100 and 2006–2035. Statistics for 30 ensembles (min, 25th percentile, median, 75th percentile, max) in each basin. Black and red lines represent the ensemble mean/individual ensemble members

Figure 2 – Were the arrow sizes in a) intended to be different sizes to indicate flux magnitude? Could you put some of your Table 1 fluxes in this figure (or derive some from the literature if you deem the simulation to be of poor quality) and size the flux arrows by relative impact? If so, you could either a) lump all basins together and create a total basin flux diagram or b) create a few of these figures with some of the more interesting groundwater basins evaluated or c) lump basins together by climate type or proximity.

Thanks for the suggestion. We have added these figure in Supplementary Information (Fig. S20 and Fig. R4 here for an example). However, each basin behaves differently, and it is hard to lump basins by the climate types.

Fig. R8 Water budget and its changes. The number is the water fluxes (mm/yr) in the current water budget, while the numbers in brackets are corresponding between two periods (2071–2100 minus 2006–2035). The sizes of an arrow are proportional to the changes (number in brackets) and blue/red for increase/decrease. The direction of the arrow represents the directions of the physical fluxes (e.g., evaporation from land to the atmosphere).

Minor Edits

I think the authors fail to convey the worth of CESM-LE in the evaluation of long-term trends to the reader. A description of why this large-ensemble approach is of great value to the scientific community needs to be conveyed. This should be expanded upon, but something like, “CESM-LE effectively samples climate internal variability in CESM through 3,000 years of simulation time which is critical when assessing decadal-to-centennial trends in slow-moving processes such as groundwater storage.”

Thanks for the suggestions. We added these discussion in *Line 74-77*.

I also think the authors fail to tell readers climate change future they chose to evaluate. In the methods, you state you used RCP8.5, but it needs to be discussed earlier on and could be described more colloquially as a “business-as-usual” scenario.

Thanks for the suggestions. We added a “business-as-usual scenario” in the abstract in *Line 33* as well as in the *Fig.3 caption*.

Line 28-29 – awkward sentence, maybe change to, “Climate change impacts on groundwater storage have the potential to compromise future water availability and sustainability.”

The sentence is reworded.

Line 30 – “...we used a fully-coupled climate model...”

The article is corrected.

Line 31 – “...critical aquifers that have been identified as significantly distressed.”

The sentence is shortened.

Line 41 – “...our results also suggest that there may be potential for enhanced groundwater use in some of the most overstressed aquifers.”

The sentence is corrected.

Line 52 – “...In terms of the hydrological cycle, climate change can affect the amount of soil infiltration due to changes in extreme precipitation, especially deeper percolation that recharge groundwater.”

The sentence is corrected.

Line 55 – “...In addition, rising temperatures increase evaporative demand over land which limits the amount of water that can replenish groundwater.”

The sentence is corrected.

Line 73 – “...provides the simulations used for this study that can be validated against the Gravity Recovery and Climate Experiment (GRACE) observations and other models.”

The sentence is corrected.

Line 83 – “..., and only account for the natural processes that act on the water balance of the aquifer.”

The sentence is rewritten.

Line 84 – remove “...and define the available supply of water.”

The sentence is rewritten together with the above.

In addition, Line 75 is an example of the overstatement of CESM’s capabilities I mentioned before. How can CESM be more realistic than GRACE, a satellite-derived product that is now a standard in the field?

Sorry for the misunderstanding. We treat GRACE as a standard as well. We clarify it in *Line 82*.

Line 91 to 93 – Although I agree that climate change will transition precipitation from snow-to-rain and deplete snowpack throughout much of the world’s mountainous regions, I don’t perceive these simulations to be credible for the assessment of snowmelt in mountainous regions.

From the snow parameterization development¹¹ in the past decades, the snow parameterization in CLM4.0 should be adequate for our current purpose of basin average. For regional or point scales, we will need higher-resolution models to resolve the snowmelt, particularly in terrain-complex regions.

Line 106 to 109 – Again, the Sierra Nevada do not resemble a real mountain range at 1-degree grid-resolution, nor do other mountain regions. Studies have shown that the maximum elevation of the Sierra Nevada at 1-degree grid-resolution is ~1600 meters which is almost 3000 meters below the actual maximum. Not only are the Sierra Nevada not resolved, but the California Central Valley is non-existent and artificially uplifted due to the Sierra Nevada being averaged into it at 1-degree resolution.

We agree that low- resolution topography dataset smooth the lower mountain peak and uplift the valley. This is not enough to present the spatial heterogeneity of snowpack in California. However, rather than looking at the spatial pattern or comparing the model results with a-single-point observation, we focus on the overall trend in a spatial average of 206,950 km² in Central Valley, where is the downstream of Sierra Nevada. The snow parametrizations in CLM4¹⁰ and low resolution should be adequate for our current purpose. We compared the topography data in CESM-LE with the one in NARCCAP, which is an international program that serves the high-resolution climate scenario of North America (Fig. R9). The maximum elevation in CESM-LE (1-degree resolution) over Central Valley is 1716 m, while the maximum elevation in CRCM+CCSM (5-km resolution) is 2605 m.

We noticed that the spatial resolution affects a lot for quantifying the snow amount from literature. We focused on the mechanisms changing groundwater storage more than giving numbers. We also incorporated the discussion of the uncertainty of the low model resolution and snow simulations in the revised main text in *Line 216-221*. A table of topography information was added to *Supplementary Information* (Table S1).

Fig. R9: (left two) Elevation (m) in CRCM-5km (right two) elevation in CAM of CESM.

Line 140 – remove “...(evaporation and transpiration).”

Removed.

Line 148-150 – awkward set of sentences.

The sentence is rewritten as:

More precipitation increases infiltration and groundwater recharge (3.7 mm/yr/dec in Table 1). Higher recharge leads to a shallower water table in the North China Plain without considering pumping. (Line 155-157)

Line 152 – Is this really the first-time groundwater storage has been assessed from a fully-coupled climate model? Why would that be the case given that you intercompared 25 different model simulation(s) in your Taylor Diagram in Supplemental Figure 1?

Based on our knowledge, previous studies³ have been used offline hydrologic models and only assessed the groundwater recharge, but not the groundwater storage. There is no groundwater model implemented into the global climate models in previous assessments. Here we first use a model that explicitly resolves groundwater storage from the GCM perspective.

In Supplemental Figure 1, TWS from CESM-LE includes the groundwater component, while other CMIP5 models do not. CESM1-WACCM, CESM1-FATCHEM, CESM1-CAM5, CESM1-BGC, and CCSM4 use the same land models. The correlations in Taylor Diagram are larger after accounting for groundwater storage (CESM-LE). Adding an unconfined aquifer beneath soil generally increases the available water capacity, which potentially enlarges the seasonal variation of total water storage (TWS).

Line 154 – “While our explanations of key mechanisms followed a more simplistic one-way approach...””

The sentence is modified.

Line 166 – “...complex interactions within the climate system...”

The sentence is modified.

Line 169 – “...changes in precipitation but also other processes module the overall evolution (e.g., evapotranspiration, snowmelt, and plant growth).

The sentence is modified.

Line 170 – “Yet, future projections in precipitation are notoriously uncertain, especially in transition zones, where the “rich-get-richer and poor-get-poorer” mechanism does hold, and where groundwater abstraction is structurally important.”

The sentence is modified.

Line 176 – “However, it is noteworthy...”

The sentence is modified.

Line 177 – “For instance, changes in timing and/or magnitude of spring snowmelt have been reported in observational data and likely influences groundwater recharge.”

The sentence is modified.

Line 181 – northwestern India is topographically complex, but how well is it resolved in CESM-LE?

The elevation used in CAM5.2 is shown in Fig. R10. The average height is 710 m; the highest elevation is 5396m. The elevation dataset can fairly represent the complex of topography.

Fig. R10: (left) Elevation (m) in CAM5.2 for resolution 0.9*1.25 (right) Elevation only for Northwestern India

Reference:

- 1 Wada, Y. *et al.* Fate of water pumped from underground and contributions to sea-level rise. *Nature Climate Change* **6**, 777, doi:10.1038/nclimate3001 (2016).
- 2 Wada, Y. *et al.* Global depletion of groundwater resources. *Geophysical Research Letters* **37**, n/a-n/a, doi:10.1029/2010GL044571 (2010).
- 3 Portmann, F., Petra, D., Stephanie, E. & Martina, F. Impact of climate change on renewable groundwater resources: assessing the benefits of avoided greenhouse gas emissions using selected CMIP5 climate projections. *Environmental Research Letters* **8**, 024023 (2013).
- 4 Schewe, J. *et al.* Multimodel assessment of water scarcity under climate change. *Proceedings of the National Academy of Sciences* **111**, 3245-3250, doi:10.1073/pnas.1222460110 (2014).
- 5 Haddeland, I. *et al.* Global water resources affected by human interventions and climate change. *Proceedings of the National Academy of Sciences* **111**, 3251-3256, doi:10.1073/pnas.1222475110 (2014).
- 6 Scanlon, B. R. *et al.* Global models underestimate large decadal declining and rising water storage trends relative to GRACE satellite data. *Proceedings of the National Academy of Sciences* (2018).
- 7 Rodell, M. *et al.* The Global Land Data Assimilation System. *Bulletin of the American Meteorological Society* **85**, 381-394, doi:10.1175/bams-85-3-381 (2004).
- 8 Anderson, E. A. A point energy and mass balance model of a snow cover. (1976).
- 9 Jordan, R. A one-dimensional temperature model for a snow cover: Technical documentation for SNTHERM. 89. (Cold Regions Research and Engineering Lab Hanover NH, 1991).
- 10 Dai, Y. & Zeng, Q. A land surface model (IAP94) for climate studies part I: Formulation and validation in off-line experiments. *Advances in Atmospheric Sciences* **14**, 433-460 (1997).

- 11 Lawrence, D. M. *et al.* Parameterization improvements and functional and structural advances in Version 4 of the Community Land Model. *Journal of Advances in Modeling Earth Systems* **3**, n/a-n/a, doi:10.1029/2011MS00045 (2011).
- 12 Niu, G. Y., Yang, Z. L., Dickinson, R. E., Gulden, L. E. & Su, H. Development of a simple groundwater model for use in climate models and evaluation with Gravity Recovery and Climate Experiment data. *Journal of Geophysical Research: Atmospheres* **112** (2007).
- 13 Kay, J. E. *et al.* The Community Earth System Model (CESM) Large Ensemble Project: A Community Resource for Studying Climate Change in the Presence of Internal Climate Variability. *Bulletin of the American Meteorological Society* **96**, 1333-1349, doi:10.1175/bams-d-13-00255.1 (2015).
- 14 Toure, A. M. *et al.* Evaluation of the Snow Simulations from the Community Land Model, Version 4 (CLM4). *Journal of Hydrometeorology* **17**, 153-170, doi:10.1175/jhmd-14-0165.1 (2016).
- 15 Zeng, Z. *et al.* Impact of Earth Greening on the Terrestrial Water Cycle. *Journal of Climate* **31**, 2633-2650, doi:10.1175/jcli-d-17-0236.1 (2018).
- 16 Fyfe, J. C. *et al.* Large near-term projected snowpack loss over the western United States. *Nature Communications* **8**, 14996, doi:10.1038/ncomms14996 (2017).
- 17 Mudryk, L., Kushner, P., Derksen, C. & Thackeray, C. Snow cover response to temperature in observational and climate model ensembles. *Geophysical Research Letters* **44**, 919-926 (2017).
- 18 Cheng, S., Huang, J., Ji, F. & Lin, L. Uncertainties of soil moisture in historical simulations and future projections. *Journal of Geophysical Research: Atmospheres* **122**, 2239-2253, doi:10.1002/2016jd025871 (2017).
- 19 Mankin, J. S., Viviroli, D., Singh, D., Hoekstra, A. Y. & Diffenbaugh, N. S. The potential for snow to supply human water demand in the present and future. *Environmental Research Letters* **10**, 114016, doi:10.1088/1748-9326/10/11/114016 (2015).
- 20 Ferguson, C. R., Pan, M. & Oki, T. The Effect of Global Warming on Future Water Availability: CMIP5 Synthesis. *Water Resources Research* **54**, 7791-7819, doi:10.1029/2018wr022792 (2018).
- 21 Oleson, K. *et al.* Improvements to the Community Land Model and their impact on the hydrological cycle. *Journal of Geophysical Research: Biogeosciences* **113** (2008).
- 22 Oleson, K. *et al.* Technical description of version 4.0 of the Community Land Model (CLM). (2010).
- 23 Gulden, L. E. *et al.* Improving land-surface model hydrology: Is an explicit aquifer model better than a deeper soil profile? *Geophysical Research Letters* **34**, doi:10.1029/2007gl029804 (2007).
- 24 Cai, X. *et al.* Assessment of simulated water balance from Noah, Noah-MP, CLM, and VIC over CONUS using the NLDAS test bed. *Journal of Geophysical Research: Atmospheres* **119**, 13,751-713,770, doi:10.1002/2014jd022113 (2014).
- 25 Lo, M.-H. & Famiglietti, J. S. Precipitation response to land subsurface hydrologic processes in atmospheric general circulation model simulations. *Journal of Geophysical Research: Atmospheres* **116**, doi:10.1029/2010JD015134 (2011).
- 26 Lin, Y.-H., Lo, M.-H. & Chou, C. Potential negative effects of groundwater dynamics on dry season convection in the Amazon River basin. *Climate Dynamics* **46**, 1001-1013, doi:10.1007/s00382-015-2628-8 (2016).

- 27 Oleson, K. *et al.* Technical description of the community land model (CLM). (2004).
- 28 Gent, P. R. *et al.* The community climate system model version 4. *Journal of Climate* **24**,
4973-4991 (2011).
- 29 Jin, J. & Wen, L. Evaluation of snowmelt simulation in the Weather Research and
Forecasting model. *Journal of Geophysical Research: Atmospheres* **117** (2012).
- 30 Rhoades, A. M., Huang, X., Ullrich, P. A. & Zarzycki, C. M. Characterizing Sierra
Nevada snowpack using variable-resolution CESM. *Journal of Applied Meteorology and
Climatology* **55**, 173-196 (2016).
- 31 Lan, C. W., Lo, M. H., Chou, C. & Kumar, S. Terrestrial water flux responses to global
warming in tropical rainforest areas. *Earth's Future* **4**, 210-224 (2016).

REVIEWERS' COMMENTS:

Reviewer #1 (Remarks to the Author):

The authors have addressed all my concerns. I have no more comments at this moment.

Yujin Zeng

Reviewer #2 (Remarks to the Author):

The extensive revisions are much appreciated. I only have minor comments left.

Major comment:

Not a major comment in terms of work, but in terms of importance: I still miss more discussion on how these results fit in with other studies, in particular offline studies (as another reviewer also mentioned). Is there a way to put the numbers from CESM into context with other numbers, so we know if they roughly agree (even just on the sign) or if they vary wildly? I'm more familiar with the climate modeling side, and less so with offline groundwater models, but I think it could really benefit the reader.

Minor comments:

Fig S3-S9: what I meant is to keep y-axis consistent per aquifer, so that the terms can be compared better per aquifer. I understand that the differences between aquifers are too large for a common y-axis.

Abstract: clarify that this study is based on a single model

L63-: it could be helpful to briefly review the sources of uncertainty in climate change-related projections, the varying degree of difficulty to quantify them, and which sources are relevant for the study here. Suggestions: (Hawkins and Sutton 2009; Lehner et al. 2020; Clark et al. 2016)

L75: I'd call this "uncertainty from internal variability", to be consistent with the literature on climate model large ensembles.

It is perfectly acceptable to concatenate historical and RCP simulations to obtain a simulation that covers the observational period. If the model was set up properly, there should be no discontinuities from 2005 to 2006.

L104-: why is this more difficult to diagnose than, say, changes in rainfall? I think most variables should be available from model output (see e.g. (Mankin et al. 2017)). I'm not saying the authors need to necessarily investigate all this, but the justification for not doing it seems weird.

L105: "region"

L107-: this sentence needs revision. What is meant with "these three feedbacks"? What feedbacks?

L122-: why?

Fig. 3: a nice and dense figure. Could the authors comment on the explained variance of Eq. (1)?

Currently, Eq. (2) enforces additivity of the three coefficients to 1 and those weighted coefficients are shown in Fig. 3, but, as can be seen in Fig. S2, there are regions where less than 80% of variance are explained. Generally, the regression model looks highly skillful, but assuming additivity everywhere is probably not justified mathematically speaking. I believe this doesn't matter hugely for the points the authors are trying to make, but I still encourage them to stress this caveat somewhere in the text. I also still think land use changes could have a (small) influence in certain regions.

L185: despite that

L190: "A" previous study?

L227-229: Suggest to tone it down a bit. Instead, maybe emphasize that these results are based on a high-emission scenario and that other, equally likely futures might hold less dramatic changes for some of the aquifers, that, however, all futures are likely to look different from the past (ref 17).

L239-: Appreciate the additional analysis, but what is even more important than robustness within one climate model, is robustness across different models. A look at the IPCC AR5 Atlas could help put the CESM1 results in broader perspective (van Oldenborgh et al. 2013).

L242: why "even though"? Considering internal variability is never a substitute to considering model uncertainty.

L248: Winter-time precipitation increases in California are probably less likely to materialize than CMIP5 suggests (Simpson et al. 2016), so this statement seems a bit too optimistic too me without further investigation of robustness across models. To that end, what is the CA Sustainable GW Act assuming will happen to future precipitation? This could be an interesting information to help connect climate model projections with on-the-ground issues as done here by the authors.

L291-292: what is the difference between natural variability and internal variability here?

References

Clark, M. P., and Coauthors, 2016: Characterizing Uncertainty of the Hydrologic Impacts of Climate Change. *Curr. Clim. Chang. Reports*, 2, 55–64, <https://doi.org/10.1007/s40641-016-0034-x>.

Hawkins, E., and R. Sutton, 2009: The potential to narrow uncertainty in regional climate predictions. *Bull. Am. Meteorol. Soc.*, 90, 1095–1107, <https://doi.org/10.1175/2009BAMS2607.1>.

Lehner, F., C. Deser, N. Maher, J. Marotzke, E. Fischer, L. Brunner, R. Knutti, and E. Hawkins, 2020: Partitioning climate projection uncertainty with multiple Large Ensembles and CMIP5/6. *Earth Syst. Dyn. Discuss.*, 1–28.

Mankin, J. S., J. E. Smerdon, B. I. Cook, A. P. Williams, and R. Seager, 2017: The curious case of projected twenty-first-century drying but greening in the American West. *J. Clim.*, 30, 8689–8710, <https://doi.org/10.1175/JCLI-D-17-0213.1>.

van Oldenborgh, G., M. Collins, J. Arblaster, J. Christensen, J. Marotzke, S. Power, M. Rummukainen, and T. Zhou, 2013: Atlas of Global and Regional Climate Projections. *Clim. Chang. 2013 Phys. Sci. Basis. Contrib. Work. Gr. I to Fifth Assess. Rep. Intergov. Panel Clim. Chang.*, 1311–1394, <https://doi.org/10.1017/CBO9781107415324.029>.

Simpson, I. R., R. Seager, M. Ting, and T. A. Shaw, 2016: Causes of change in Northern Hemisphere winter meridional winds and regional hydroclimate. *Nat. Clim. Chang.*, 6, 65–70, <https://doi.org/10.1038/nclimate2783>.

Reviewer #3 (Remarks to the Author):

Summary

Wu et al. have provided substantial revisions to the manuscript, "Divergent effects of climate change on future groundwater availability in key mid-latitude aquifers." It is readily apparent that a meticulous approach to my (and other reviewer's) comments were taken, which is very much appreciated. As mentioned in my previous summary comment, I think this manuscript has a lot of merit and my comments (although significant) were merely aimed at ensuring that all possible uncertainties were addressed as I think this could become a widely cited study (and useful for climate change assessments). I have a few more suggested minor revisions and comments and would advise that the editor accept once these are addressed.

Reply to Major Revision Comments

Reply to comment 1) – Thank you for now including groundwater pumping estimates (using the Wada et al. methods) to juxtapose natural variability and anthropogenic climate change induced trends to groundwater storage simulated in the CESM-LENS. It is very interesting (and makes sense based on current pump log and GRACE observations) that positive groundwater recharge induced by natural climate variability is overwhelmed by the negative pumping induced trends in both the Central Valley and Northwestern India. As an FYI, in the Figure 4 caption the error bar symbol (below 21st c. climate w/o pumping text) is shown but has no caption. I think this should be deleted. Also, FWIW, Figure S2 is a very nice addition to this manuscript.

Reply to comment 2) – Thank you for comprehensively addressing this comment throughout the text. I think the authors do a much better job of caveating the CESM-LENS findings and setting CESM's performance in the context of other Earth system and groundwater hydrology models. On that note, I'm glad to see that Reviewer #2's comment regarding the interpretation of the Figure S1 Taylor Diagram – CSIRO-Mk3.6.0 is the "best model" when compared with GRACE observations – has been addressed.

Reply to comment 3) – I'm a bit unclear how the GLDAS-Noah 0.25-degree model produces equivalently small SWE estimates as the 1-degree model in the time series (i.e., $<25 \text{ kg/m}^2$ [or 25 mm or ~ 1 inch]). Is this because the authors use basin-averages over the entire region shown? If so, a lot of "0" SWE values will dampen down the Sierra Nevada "signal" of SWE and make the comparison a bit misleading. Also, GLDAS-Noah would have no bearing (very different choices in snow model structure and parameter choices between Noah and CLM) on the authors results in this study (and I'm unclear why the example is used). With that said, my whole point with this comment was not to dispute CLM4.0's snowpack process representation. CLM has been the foundation of much of my work and I'm well acquainted with its sophistication relative to other Earth system models (and even regional climate models). Rather, my point was to bring up the fact that certain mountain ranges (e.g., California Sierra Nevada, and others) are not adequately resolved at 1-degree and will not induce proper orographic uplift on storm events (total precipitation) and proper rain-snow partitioning (snowfall totals) and thus not accurately represent the accumulation (or melt) dynamics of the seasonal snowpack. Therefore, given the aforementioned, I still believe it is hard to deduce groundwater recharge/trends in snowmelt dominated basins in the CESM-LENS simulations, at least for certain mountain ranges. As mentioned in the GLDAS-Noah comparison, using basin averages can dampen the "signal" of snowpack representation and overall melt dynamics that are important to both runoff and groundwater recharge totals from the headwaters to valleys of basins chosen in this study. For

example, this point is apparent in Figure 3 of the revised manuscript where snowmelt is not the dominant driver of groundwater recharge across most of the western US even though it is widely accepted in the literature as being so (e.g., Li et al., 2017 - <https://agupubs.onlinelibrary.wiley.com/doi/10.1002/2017GL073551>). I think the authors now caveat their findings somewhat in the new manuscript, but (given the aforementioned) needs a bit more recognition in the Results and/or Discussion section(s). I also wonder if it's feasible to evaluate how the basins (particularly snowmelt dominated ones) in Figure 3 (revised manuscript) might change with a 0.25 degree resolution simulation vs 1.00 degree resolution simulation. FWIW, Figure 3 is really neat!

Reply to comment 4) – I truly hope that the authors pursue this topic in a future manuscript (i.e., intersection of changes in soil characteristics, pre- and post-drought soil moisture content, and mean vs extreme precipitation changes on groundwater recharge). Also, in Fig. R3, the changes in soil water content (SWC) in Northwestern India are surprisingly small. I'm not sure what the confidence intervals are on these estimates, but it seems historical vs future long-term averages would be nearly the same across all soil levels? Is this expected in a radically changing climate? Or, is this a function of the structural uncertainty of groundwater representation in CLM (i.e., bucket model approach)?

Reply to Minor Revision Comments

Reply to Figure 1 (new manuscript) revision requests – You may want to highlight in the Figure 1 caption that a basin-by-basin breakdown across the 7 groundwater basins is shown in Figure S20. Hopefully this will “flag” readers to read the supplemental more carefully. My anticipation is that many of the supplemental figures may get “buried” and lost to the casual reader (an unfortunate outcome for the amount of work done in the revisions).

Reply to Table 1 – Can you put error bars on all these trend estimates in the main manuscript (i.e., 95% confidence intervals across the 30 CESM-LENS members)? This seems to have been done for Table S2 (not sure why it was done there and not here as well).

Line (after) 88-90 – I still think the authors should be upfront here and provide a sentence or two regarding the processes that shape groundwater in CESM that are oversimplifications and how they might influence the changes/trends mentioned in the results (particularly by end-century). While the authors state this further in the manuscript (i.e., Discussion), it might be helpful for readers to know these limitations earlier on (see “Reply to comment 4”) which provides an example of potential structural uncertainty in CLM groundwater representation).

Figure S10 – There is yellow highlighted text in the caption.

Figure S20 – It might be helpful to fill the lower right (or upper left) hand corner of the plot with a global map plot of the regions analyzed in this study (similar to Figure 2), particularly since there is empty space available and would make this figure standalone.

Replies to reviewer's comments for manuscript NCOMMS-18-07827A
Divergent effects of climate change on future groundwater availability in key mid-latitude aquifers

Reviewer's original comments are in black.

The responses are in blue. Line number is corresponding to new and non-track version.

Dear reviewers,

Thanks very much for your advice on manuscript NCOMMS-18-07827A submitted to Nature Communications for publication. We have taken care of all the reviewers' questions. In the following, please find our detailed response to your comments and the revised manuscript.

Reviewer #2 (Remarks to the Author):

The extensive revisions are much appreciated. I only have minor comments left.

Major comment:

Not a major comment in terms of work, but in terms of importance: I still miss more discussion on how these results fit in with other studies, in particular offline studies (as another reviewer also mentioned). Is there a way to put the numbers from CESM into context with other numbers, so we know if they roughly agree (even just on the sign) or if they vary wildly? I'm more familiar with the climate modeling side, and less so with offline groundwater models, but I think it could really benefit the reader.

Thank you for the great comments. Taylor et al.¹ indicated that our understanding of the impact of climate change on groundwater is still limited. Previous studies of groundwater changes under climate change are based on offline (standalone) hydrological models or regional groundwater models. The global-scale studies using offline hydrological models mostly focus on groundwater recharge changes. Also, most of them do not include a geographical analysis, such as zoom in to a groundwater aquifer and analyze the water budget. Our results (the sign of groundwater recharge changes) agree with Döll² and Portmann et al.³ in the Southern Plains(-), Northwestern India(+), the North China Plain(+). It is determined by the global map shown in their papers.

Meixner et al.⁴ reviewed multiple studies of using groundwater models to investigate the impact of climate change on groundwater recharge in the US by 2100. They concluded that groundwater recharge decreases by 5% with large uncertainties of 25% in the Central Valley, while our results show the groundwater storage has no significant trends ($1.8 \pm 3.3 \text{ mm dec}^{-1}$). They also concluded that groundwater recharge decreases by 10% in the Southern Great Plains, which is similar to our

result of decreasing groundwater storage with a rate of -23.3mm dec^{-1} . We add the similarity discussion to previous studies in the Southern Plains in *Line 118* of the revised manuscript. Also, the above discussion is in Supplementary Discussion.

Minor comments:

Fig S3-S9: what I meant is to keep y-axis consistent per aquifer, so that the terms can be compared better per aquifer. I understand that the differences between aquifers are too large for a common y-axis.

Thanks for the comment. The changes in the seasonal cycle are often not distinguishable using the same y-axis per aquifer (Supplementary Figures 3-9) because of the large range in the y-axis. Supplementary Figure 20 can show the concept that the reviewer mentioned, which is the comparison between different fluxes in the water budget.

Abstract: clarify that this study is based on a single model

Yes. It has been mentioned in *Line 25* in the revised manuscript. *“Here, we used a fully-coupled climate model to investigate changes in GWS over seven critical aquifers.”*

L63-: it could be helpful to briefly review the sources of uncertainty in climate change-related projections, the varying degree of difficulty to quantify them, and which sources are relevant for the study here. Suggestions: (Hawkins and Sutton 2009; Lehner et al. 2020; Clark et al. 2016)

We add a brief statement for uncertainties in climate projections in *Line 55* in the revised manuscript.

L75: I'd call this “uncertainty from internal variability”, to be consistent with the literature on climate model large ensembles.

The sentence is corrected.

It is perfectly acceptable to concatenate historical and RCP simulations to obtain a simulation that covers the observational period. If the model was set up properly, there should be no discontinuities from 2005 to 2006.

Yes, the time series shows no discontinuities. We guess that your comment is for Fig.S1. We think the historical and RCP should be separated for evaluating the model to distinguish the uncertainties from greenhouse gas projections.

L104-: why is this more difficult to diagnose than, say, changes in rainfall? I think most variables should be available from model output (see e.g. (Mankin et al. 2017)). I'm not saying the authors need to necessarily investigate all this, but the justification for not doing it seems weird.

To avoid confusion, we removed the sentence. The discussion of phenology could be found in Supplementary Discussion 3.

L105: "region"

It is corrected.

L107-: this sentence needs revision. What is meant with "these three feedbacks"? What feedbacks?

The sentence is revised as the following in the revised manuscript. "*Rainfall, ET, and snowmelt are three factors act as the primary mechanisms for driving GWS changes (Fig. 1).*"

L122-: why?

We assume your question is related to the seasonality of the groundwater recharge and the soil water. Human adapts to the seasonality of the water supply to make production more efficiently—for example, the growing season for rice paddy. Changes in the seasonality (timing) of groundwater recharge will make an influence on pumping behavior (timing of water demand).

Fig. 3: a nice and dense figure. Could the authors comment on the explained variance of Eq. (1)? Currently, Eq. (2) enforces additivity of the three coefficients to 1 and those weighted coefficients are shown in Fig. 3, but, as can be seen in Fig. S2, there are regions where less than 80% of variance are explained. Generally, the regression model looks highly skillful, but assuming additivity everywhere is probably not justified mathematically speaking. I believe this doesn't matter hugely for the points the authors are trying to make, but I still encourage them to stress this caveat somewhere in the text. I also still think land use changes could have a (small) influence in certain regions.

As the reviewer mentioned, the multiple linear regression method is powerful but not perfect. The variance could be the interactions between the predictors, non-linear effects, and the time-lagged effects. We use an 11-year running mean here, but for some arid regions, the residence time could be longer. Other factors may also play a role but are hard to quantify using the regression (e.g., land cover change, soil property change, frozen soil, frequency/intensity of

precipitation, as Supplementary Discussion 2). We hypothesize that the regression model does not work well in the desert region where groundwater recharge is minimal. We also masked out the glacier region and did not show the polar region in Fig. 3.

L185: despite that

The sentence is modified.

L190: “A” previous study?

The article is added.

L227-229: Suggest to tone it down a bit. Instead, maybe emphasize that these results are based on a high-emission scenario and that other, equally likely futures might hold less dramatic changes for some of the aquifers, that, however, all futures are likely to look different from the past (ref 17).

Thank you for the suggestion. Now we indicate that the result is model-dependent and scenario-dependent in *Line 238* “*our results may remain model-dependent and scenario-dependent*”.

L239-: Appreciate the additional analysis, but what is even more important than robustness within one climate model, is robustness across different models. A look at the IPCC AR5 Atlas could help put the CESM1 results in broader perspective (van Oldenborgh et al. 2013).

We analyzed the robustness in groundwater trends due to climate change by comparing it to the pre-industrial experiments. However, most of the GCMs in CMIP5 do not model groundwater. Also, groundwater-related variables are not included in the CMIP5 archive.

L242: why “even though”? Considering internal variability is never a substitute to considering model uncertainty.

We agree that internal variability is different from the model uncertainty. Studies have found that the spread in CESM-LE is comparable to models of the Coupled Model Intercomparison Project phase 5 (CMIP5) in ENSO amplitude⁵ and temperature trends⁶. The statement in the main text is removed to avoid confusion.

L248: Winter-time precipitation increases in California are probably less likely to materialize than CMIP5 suggests (Simpson et al. 2016), so this statement seems a bit too optimistic to me without further investigation of robustness across models. To that end, what is the CA Sustainable GW Act assuming will happen to future precipitation? This could be an interesting

information to help connect climate model projections with on-the-ground issues as done here by the authors.

As reviewers mentioned, several studies^{7,8} suggest that most GCMs overestimate the meridional winds and therefore overestimate the winter precipitation in California. Our results show that groundwater storage in the Central Valley slightly increases and it is not statistically significant. Therefore, we removed Central Valley in this statement.

Furthermore, California's Sustainable Groundwater Management Act (SGMA) implicitly includes the climate change concept. SGMA requires groundwater management that should have climate change assessment and implement 50-year planning. On the other hand, no specific framework for setting this threshold is officially required to use in SGMA. The recommended datasets show slight increases in precipitation by 2.9% in 2030 and 5.3% in 2070, based on the GCMs outputs with bias correction⁹. More importantly, the partitioning of precipitation into rainfall and snowfall changes. It indicated that snowfall and snowmelt reduction decreases the groundwater recharge and causes less groundwater for pumping⁹. The major driver in the SGMA is withdrawal and overdrafting, not recharge. Therefore, even if the recharge increases slightly, the pumping is much greater and would still have to be reduced. Compared to the current pumping rate (Fig. 4), we think precipitation changes are a minor issue for setting this threshold.

.

L291-292: what is the difference between natural variability and internal variability here?

There are no differences, though it could be slightly different, depending on the context (e.g., whether including natural forcings such as solar activity and volcanic activity).

The sentence is rewritten as follow:

“Since the anthropogenic pumping was not simulated by CESM-LE, the long-term GWS changes can be fully attributed to anthropogenic climate change and internal climate variability.”

References

Clark, M. P., and Coauthors, 2016: Characterizing Uncertainty of the Hydrologic Impacts of Climate Change. *Curr. Clim. Chang. Reports*, 2, 55–64, <https://doi.org/10.1007/s40641-016-0034-x>.

Hawkins, E., and R. Sutton, 2009: The potential to narrow uncertainty in regional climate predictions. *Bull. Am. Meteorol. Soc.*, 90, 1095–1107, <https://doi.org/10.1175/2009BAMS2607.1>.

Lehner, F., C. Deser, N. Maher, J. Marotzke, E. Fischer, L. Brunner, R. Knutti, and E. Hawkins, 2020: Partitioning climate projection uncertainty with multiple Large Ensembles and CMIP5/6. *Earth Syst. Dyn. Discuss.*, 1–28.

Mankin, J. S., J. E. Smerdon, B. I. Cook, A. P. Williams, and R. Seager, 2017: The curious case of projected twenty-first-century drying but greening in the American West. *J. Clim.*, 30, 8689–8710, <https://doi.org/10.1175/JCLI-D-17-0213.1>.

van Oldenborgh, G., M. Collins, J. Arblaster, J. Christensen, J. Marotzke, S. Power, M. Rummukainen, and T. Zhou, 2013: Atlas of Global and Regional Climate Projections. *Clim. Chang. 2013 Phys. Sci. Basis. Contrib. Work. Gr. I to Fifth Assess. Rep. Intergov. Panel Clim. Chang.*, 1311–1394, <https://doi.org/10.1017/CBO9781107415324.029>.

Simpson, I. R., R. Seager, M. Ting, and T. A. Shaw, 2016: Causes of change in Northern Hemisphere winter meridional winds and regional hydroclimate. *Nat. Clim. Chang.*, 6, 65–70, <https://doi.org/10.1038/nclimate2783>.

Reviewer #3 (Remarks to the Author):

Summary

Wu et al. have provided substantial revisions to the manuscript, “Divergent effects of climate change on future groundwater availability in key mid-latitude aquifers.” It is readily apparent that a meticulous approach to my (and other reviewer’s) comments were taken, which is very much appreciated. As mentioned in my previous summary comment, I think this manuscript has a lot of merit and my comments (although significant) were merely aimed at ensuring that all possible uncertainties were addressed as I think this could become a widely cited study (and useful for climate change assessments). I have a few more suggested minor revisions and comments and would advise that the editor accept once these are addressed.

Reply to Major Revision Comments

Reply to comment 1) – Thank you for now including groundwater pumping estimates (using the Wada et al. methods) to juxtapose natural variability and anthropogenic climate change induced trends to groundwater storage simulated in the CESM-LENS. It is very interesting (and makes sense based on current pump log and GRACE observations) that positive groundwater recharge induced by natural climate variability is overwhelmed by the negative pumping induced trends in both the Central Valley and Northwestern India. As an FYI, in the Figure 4 caption the error bar symbol (below 21st c. climate w/o pumping text) is shown but has no caption. I think this should be deleted. Also, FWIW, Figure S2 is a very nice addition to this manuscript.

The error bar in the legend is deleted in the revised manuscript.

Reply to comment 2) – Thank you for comprehensively addressing this comment throughout the text. I think the authors do a much better job of caveating the CESM-LENS findings and setting CESM’s performance in the context of other Earth system and groundwater hydrology models. On that note, I’m glad to see that Reviewer #2’s comment regarding the interpretation of the Figure S1 Taylor Diagram – CSIRO-Mk3.6.0 is the “best model” when compared with GRACE observations – has been addressed.

Thank you for the comment.

Reply to comment 3) – I’m a bit unclear how the GLDAS-Noah 0.25-degree model produces equivalently small SWE estimates as the 1-degree model in the time series (i.e., $<25 \text{ kg/m}^2$ [or 25 mm or ~1 inch]). Is this because the authors use basin-averages over the entire region shown? If so, a lot of “0” SWE values will dampen down the Sierra Nevada “signal” of SWE and make the comparison a bit misleading. Also, GLDAS-Noah would have no bearing (very different choices in snow model structure and parameter choices between Noah and CLM) on the authors results in this study (and I’m unclear why the example is used). With that said, my whole point with this comment was not to dispute CLM4.0’s snowpack process representation. CLM has been the foundation of much of my work and I’m well acquainted with its sophistication relative to other Earth system models (and even regional climate models). Rather, my point was to bring up the fact that certain mountain ranges (e.g., California Sierra Nevada, and others) are not adequately resolved at 1-degree and will not induce proper orographic uplift on storm events (total precipitation) and proper rain-snow partitioning (snowfall totals) and thus not accurately represent the accumulation (or melt) dynamics of the seasonal snowpack. Therefore, given the aforementioned, I still believe it is hard to deduce groundwater recharge/trends in snowmelt dominated basins in the CESM-LENS simulations, at least for certain mountain ranges. As mentioned in the GLDAS-Noah comparison, using basin averages can dampen the “signal” of snowpack representation and overall melt dynamics that are important to both runoff and groundwater recharge totals from the headwaters to valleys of basins chosen in this study. For example, this point is apparent in Figure 3 of the revised manuscript where snowmelt is not the dominant driver of groundwater recharge across most of the western US even though it is widely accepted in the literature as being so (e.g., Li et al., 2017 - <https://agupubs.onlinelibrary.wiley.com/doi/10.1002/2017GL073551>). I think the authors now caveat their findings somewhat in the new manuscript, but (given the aforementioned) needs a bit more recognition in the Results and/or Discussion section(s). I also wonder if it’s feasible to evaluate how the basins (particularly snowmelt dominated ones) in Figure 3 (revised manuscript) might change with a 0.25 degree resolution simulation vs 1.00 degree resolution simulation. FWIW, Figure 3 is really neat!

We agree that GLDAS-Noah is not a perfect example. We intended to show that the coarser-resolution causes an underestimation of snow quantities, but the sign of the trends is not affected. Rainfall is often the dominant factor in Fig.3. However, compared to other regions, snowmelt contribution in the western US pops up in Fig.S11c (as shown below R1). The results shown in this study might underestimate the contribution of snowmelt.

We agree that it will be interesting to see how Fig. 3 changes with resolution. We hypothesize that the contribution of snowmelt in headwater regions would be larger with finer resolution. It is an interesting topic and is doable at a continental scale.

Figure R1 (Supplementary Figure 11 c.) Contribution of snowmelt.

Reply to comment 4) – I truly hope that the authors pursue this topic in a future manuscript (i.e., intersection of changes in soil characteristics, pre- and post-drought soil moisture content, and mean vs extreme precipitation changes on groundwater recharge). Also, in Fig. R3, the changes in soil water content (SWC) in Northwestern India are surprisingly small. I'm not sure what the confidence intervals are on these estimates, but it seems historical vs future long-term averages would be nearly the same across all soil levels? Is this expected in a radically changing climate? Or, is this a function of the structural uncertainty of groundwater representation in CLM (i.e., bucket model approach)?

Thank you for the encouragement and suggested topic, which can be a very interesting future work.

In previous Fig. R3 (Fig. R2a below), the changes in SWC is small. But during summer, the variation in the top SWC could be ~ -0.02 , which is not a small change compared to the mean SWC of 0.17. We also see the different changes in surface/deep soil moisture in some seasons (Fig. R2b below). Soil gets drier especially during the monsoon (June and July) and gets wetter briefly in November. The drying of soil moisture is more obvious at the surface, while the bottom layers get wetter.

Figure R2. (a) (previous Fig. R3) Long-term average soil moisture profile over Northwestern India. Black and red lines represent the ensemble mean during 2006–2035 (more recent) and 2071–2100 (future), respectively. The figure shows a drying above 1.5 m, and wetting below it. The modeled water table depth in the regions is below 5 m. (b) changes in seasonality of soil profile (future-recent)

Reply to Minor Revision Comments

Reply to Figure 1 (new manuscript) revision requests – You may want to highlight in the Figure 1 caption that a basin-by-basin breakdown across the 7 groundwater basins is shown in Figure S20. Hopefully this will “flag” readers to read the supplemental more carefully. My anticipation is that many of the supplemental figures may get “buried” and lost to the casual reader (an unfortunate outcome for the amount of work done in the revisions).

Thank you for the suggestion. The linkage to Supplementary Figure 20 is added.

Reply to Table 1 – Can you put error bars on all these trend estimates in the main manuscript (i.e., 95% confidence intervals across the 30 CESM-LENS members)? This seems to have been done for Table S2 (not sure why it was done there and not here as well).

Yes, we replaced Table 1 with Supplementary Table 2.

Line (after) 88-90 – I still think the authors should be upfront here and provide a sentence or two regarding the processes that shape groundwater in CESM that are oversimplifications and how they might influence the changes/trends mentioned in the results (particularly by end-century). While the authors state this further in the manuscript (i.e., Discussion), it might be helpful for

readers to know these limitations earlier on (see “Reply to comment 4”) which provides an example of potential structural uncertainty in CLM groundwater representation).

This is noted in the revised manuscript: “*It should be noted that the simple groundwater model in CLM4.0 has its limitations at local-scale.*”

Figure S10 – There is yellow highlighted text in the caption.

The format is corrected.

Figure S20 – It might be helpful to fill the lower right (or upper left) hand corner of the plot with a global map plot of the regions analyzed in this study (similar to Figure 2), particularly since there is empty space available and would make this figure standalone.

The geographic legend is shown.

Reference

- 1 Taylor, R. G. *et al.* Ground water and climate change. *Nature Clim. Change* **3**, 322-329 (2013).
- 2 Döll, P. Vulnerability to the impact of climate change on renewable groundwater resources: a global-scale assessment. *Environmental Research Letters* **4**, 035006 (2009).
- 3 Portmann, F., Petra, D., Stephanie, E. & Martina, F. Impact of climate change on renewable groundwater resources: assessing the benefits of avoided greenhouse gas emissions using selected CMIP5 climate projections. *Environmental Research Letters* **8**, 024023 (2013).
- 4 Meixner, T. *et al.* Implications of projected climate change for groundwater recharge in the western United States. *Journal of Hydrology* **534**, 124-138 (2016).
- 5 Zheng, X.-T., Hui, C. & Yeh, S.-W. Response of ENSO amplitude to global warming in CESM large ensemble: uncertainty due to internal variability. *Climate Dynamics* **50**, 4019-4035 (2018).
- 6 Kay, J. E. *et al.* The Community Earth System Model (CESM) Large Ensemble Project: A Community Resource for Studying Climate Change in the Presence of Internal Climate Variability. *Bulletin of the American Meteorological Society* **96**, 1333-1349 (2015).
- 7 Simpson, I. R., Seager, R., Ting, M. & Shaw, T. A. Causes of change in Northern Hemisphere winter meridional winds and regional hydroclimate. *Nature Climate Change* **6**, 65-70 (2016).
- 8 Hagos, S. M., Leung, L. R., Yoon, J.-H., Lu, J. & Gao, Y. A projection of changes in landfalling atmospheric river frequency and extreme precipitation over western North America from the Large Ensemble CESM simulations. *Geophysical Research Letters* **43**, 1357-1363 (2016).
- 9 California Department of Water Resources, Guidance for Climate Change Data Use During Groundwater Sustainability Plan Development (2018)
<https://data.cnra.ca.gov/dataset/sgma-climate-change-resources/resource/f824eb68-1751-4f37-9a15-d9edbc854e1f>